# A truncated variant of the ribosome-associated trigger factor specifically contributes to plant chloroplast ribosome biogenesis

Fabian Ries [1,10], Jasmin Gorlt [1], Sabrina Kaiser [2], Vanessa Scherer[3], Charlotte Seydel [4], Sandra Nguyen[1], Andreas Klingl [4], Julia Legen[5], Christian Schmitz-Linneweber[5], Hinrik Plaggenborg [6], Jediael Z. Y. Ng [7], Dennis Wiens[7], Georg K. A. Hochberg [7,8], Markus Räschle [9], Torsten Möhlmann [3], David Scheuring [2] & Felix Willmund [1,11] ✉

Molecular chaperones are essential throughout a protein's life and act already during protein synthesis. Bacteria and chloroplasts of plant cells share the ribosome-associated chaperone trigger factor (Tig1 in plastids), facilitating maturation of emerging nascent polypeptides. While typical trigger factor chaperones employ three domains for their task, the here described truncated form, Tig2, contains just the ribosome binding domain. Tig2 is widely present in green plants and appears to have acquired an entirely different task than co-translational nascent polypeptide folding. Tig2 deletion results in remarkable leaf developmental defects of cold-exposed *Arabidopsis thaliana* plants and specific defects in plastidic ribosomes. Our data indicate that Tig2 functions during ribosome biogenesis by promoting the maturation of the large subunit. We hypothesize that Tig2 binding to the ribosomal tunnel-exit surface aids protecting this sensitive surface during assembly. Tig2 illustrates a fascinating concept of how a chaperone domain evolved individually, serving a completely different molecular task.

The semi-autonomous chloroplast genome (plastome) encodes approximately 100 proteins, many of which are essential subunits of the photosynthetic machinery[1]. Chloroplast gene expression is mainly controlled by post-transcriptional processes, including protein synthesis[2–4]. Such control is crucial for the efficient adjustment of the photosynthesis machinery in response to changing environmental conditions. Furthermore, co-translational regulation orchestrates the assembly of nucleus- and chloroplast-encoded subunits into common complexes, as seen for photosystem I and II (PSI and PSII), the cytochrome $b_6f$ complex, the ATP synthase or the Ribulose-1,5-Bisphosphate-Carboxylase/Oxygenase (RuBisCO). The ribosome itself is composed of subunits of dual origins with approximately half of the

[1]Molecular Genetics of Eukaryotes, University of Kaiserslautern, Kaiserslautern, Germany. [2]Plant Pathology, University of Kaiserslautern, Kaiserslautern, Germany. [3]Plant Physiology, University of Kaiserslautern, Kaiserslautern, Germany. [4]Plant Development, Ludwig-Maximilians-University Munich, Planegg-Martinsried, Germany. [5]Molecular Genetics, Humboldt-University of Berlin, Berlin, Germany. [6]Molecular Plant Sciences & Synmikro, University of Marburg, Marburg, Germany. [7]Max-Planck-Institute for Terrestrial Microbiology, Marburg, Germany. [8]Evolution Biology & Synmikro, University of Marburg, Marburg, Germany. [9]Molecular Genetics, University of Kaiserslautern, Kaiserslautern, Germany. [10]Present address: Institute of Systems Biotechnology, Saarland University, Saarbrücken, Germany. [11]Present address: Molecular Plant Sciences & Synmikro, University of Marburg, Marburg, Germany. ✉e-mail: willmund@staff.uni-marburg.de

ribosomal proteins of the small 30S ribosomal subunit being encoded by the plastome or the nuclear genome, respectively. In contrast, a quarter of the proteins of the large 50S are plastid-encoded[5]. Ribosome assembly is a complex process that involves at least 30 assembly factors in bacterial cells and more than 250 factors in the cytosol of eukaryotes[6,7]. Several factors involved in chloroplast ribosome assembly have been described, but the underlying mechanisms remain poorly understood (reviewed in ref. [8]). It is noteworthy that not all assembly factors have homologs in bacteria. This suggests that the assembly of the plastid translation machinery is more sophisticated than its prokaryotic counterpart.

Comparable to plastid gene expression, the folding and maturation of newly synthesized proteins in the chloroplast is achieved by factors that are homologous to the bacterial protein homeostasis machinery. However, important mechanistic differences evolved after the endosymbiotic event[9,10]. In bacteria and chloroplasts, the initial stage of protein maturation is promoted by the ribosome-associated and ATP-independent molecular chaperone, designated trigger factor[10–13]. With its distinctive dragon-shaped conformation, trigger factor arches over the ribosome-exit tunnel for binding of the emerging nascent polypeptides and for preventing unwanted interactions with other cellular proteins[13]. The chloroplast trigger factor, designated Tig1, exhibits a comparable overall architecture to its extensively studied bacterial homolog. However, the charge distribution of the amino acids within the chaperone domain and the arrangements of the individual domains is notably altered in the chloroplast chaperone, which supports its specialized role in chloroplast protein biogenesis[11,12,14]. The deletion of the chaperone is non-lethal in *Chlamydomonas reinhardtii* and *Arabidopsis thaliana* (Arabidopsis thereafter), yet it results in an altered energy homeostasis as indicated by an increased ratio of cyclic over linear electron flow and increased energy demand of mutant algae. Furthermore, mutants exhibit an increased occurrence of chloroplast polysomes, which may suggest that misfolding is compensated by increased biosynthesis[12].

During our analysis of the Tig1 function in chloroplasts, a completely uncharacterized gene, encoding a second putative chloroplast trigger factor-like protein, termed Tig2, sparked our interest. Tig2 is a conserved protein in streptophytes and has been identified in chloroplast stromal mega-Dalton complexes by proteomic analysis[11,15]. We now studied why green plants express two trigger factor forms and if the truncated Tig2 is conducting a similar function than Tig1. We show that Tig2 lacks chaperone activity but dominantly associates with ribosomes. The deletion of Tig2 results in a cold-sensitive phenotype, characterized by chlorotic young leaves and reduced photosynthesis rates. Further analyses of the Arabidopsis *tig2* mutant indicates that chloroplast ribosome biogenesis is impaired, which is specifically manifested by impaired hidden rRNA break formation near the ribosome tunnel exit site. Hidden breaks are introduced by rRNA cleavage in a post-maturation step of ribosome biogenesis. Our data indicate that this truncated chaperone evolved specifically to protect the ribosome exit site during ribosome maturation.

## Results

### Features of Trigger factor 2 proteins

Full-length trigger factor is found in prokaryotes and the genomes of photosynthetic eukaryotes (Tig1), where it localizes to chloroplasts. A second gene, encoding a truncated trigger factor protein (Tig2), was previously identified in all land plants (Embryophyta) investigated, but not in the initially studied algal species or prokaryotes[11]. Here, a more comprehensive phylogenetic screen of Tig1 and Tig2 trigger factor sequences revealed that Tig2 sequences appear to be absent in cyanobacteria and present in only a few chlorophyte species, whereas they are frequently present in streptophytes. This suggests that a duplication of the *Tig* gene occurred before the divergence of chlorophytes and streptophytes, giving rise to Tig1 and Tig2, and that Tig2 was

subsequently lost in most chlorophytes (Fig. 1a and Supplementary Fig. 1). The full-length trigger factor sequences that diverged before this duplication should therefore be classified separately, as they are derived from a trigger factor homolog that is ancestral to both *Tig1* and *Tig2*. The branch leading the Tig2 clade is much longer than the one leading to the Tig1 clade. This is consistent with the observation that Tig1 retains broadly the same function as prokaryotic trigger factor, whereas Tig2 may have acquired a novel function.

In Arabidopsis, mature Tig2 (At2g30695, minus the 54 N-terminal amino acids of the chloroplast transit sequence) exclusively aligns with the N-terminal ribosome binding domain of *E. coli* trigger factor (*Ec*TF) or Arabidopsis Tig1. Tig2 terminates just before the start of the flexible linker region of Tig1, which connects the N-terminal domain and the peptidyl-prolyl *cis/trans* isomerization domain (Fig. 1b, c). The ribosome binding domains of Arabidopsis Tig1 and Tig2 share only ~15% amino acid identity and 34.5% similarity, respectively. However, the AlphaFold[16,17] predicted structural conformations show striking similarities between the two molecules with a bilateral architecture. On one side, a positively charged, protruding helix-loop-helix fold contains the GFR ribosome-binding motif within a typical loop that is responsible for ribosome docking[18]. The loop appears to be more extended for Tig2, when compared to Tig1. The opposite side of the molecule is shaped by four predicted beta strands (Fig. 1d and Supplementary Fig. 2). These models indicate that Tig2 indeed has a bona fide trigger factor-like conformation. To ensure that the AlphaFold model of Tig1 is reliable, we compared the conformation to our previous SAXS data (Supplementary Fig. 2c) and the crystal structure of the truncated Chlamydomonas Tig1 molecule[11,14]. The domain architecture appears overall reliably predicted by AlphaFold, while orientation of the domains might be slightly altered.

The predicted secondary structure content was validated by far-UV circular dichroism (CD) spectroscopy comparison with heterologously expressed and purified Arabidopsis Tig1 and Tig2 proteins (Fig. 1e). The spectra of both proteins were comparable, with pronounced minima at 208 and 222 nm, which are indicative of proteins with a high α-helical secondary structural content (Fig. 1f, left panel). Thermal unfolding was examined by gradual heating and CD spectroscopy at 208 and 222 nm. Tig1 and Tig2 exhibited thermolabile behavior above 30 °C and unfolding midpoints between 40 °C and 45 °C (Fig. 1f, right panel). This is consistent with our previously described heat-labile and aggregation-prone features of Tig1, even under moderate heat stress[11].

The capacity of Tig2 to act as a chaperone was evaluated by examining its ability to prevent aggregation of the two model substrates RbcL (the large subunit of the *Chlamydomonas* RuBisCO) and GAPDH (glyceraldehyde-3-phosphate dehydrogenase). Both proteins are unable to undergo intrinsic refolding upon chemical denaturation and served as suitable model substrates for *Ec*TF and Tig1 chaperone activity[12,19,20]. In the absence of chaperone, dynamic light scattering (DLS) demonstrates that the unfolded RbcL and GAPDH rapidly aggregate if diluted in folding buffer (Fig. 1g and Supplementary Fig. 3)[21]. The presence of a ten to fiftyfold molar excess of Tig1 was found to reduce aggregation of RbcL and GAPDH (Supplementary Fig. 3B). However, any tested concentration of Tig2 failed to reduce aggregation (Fig. 1g, right panel). Consequently, Tig2 is unable to act as aggregate-preventing chaperone, at least for our assay with the two tested model substrates GAPDH and RbcL. This is in accordance with the absence of the major C-terminal chaperone module of bona fide trigger factor molecules, which is required for binding of unfolded protein[13].

### Specific expression of Tig2

For subcellular localization, the transit peptide of Tig2 (Fig. 1b) is unambiguously predicted for chloroplast translocation. To experimentally confirm this, transgenic Arabidopsis lines were generated, which stably express the Tig1 or Tig2 proteins with a C-terminal GFP

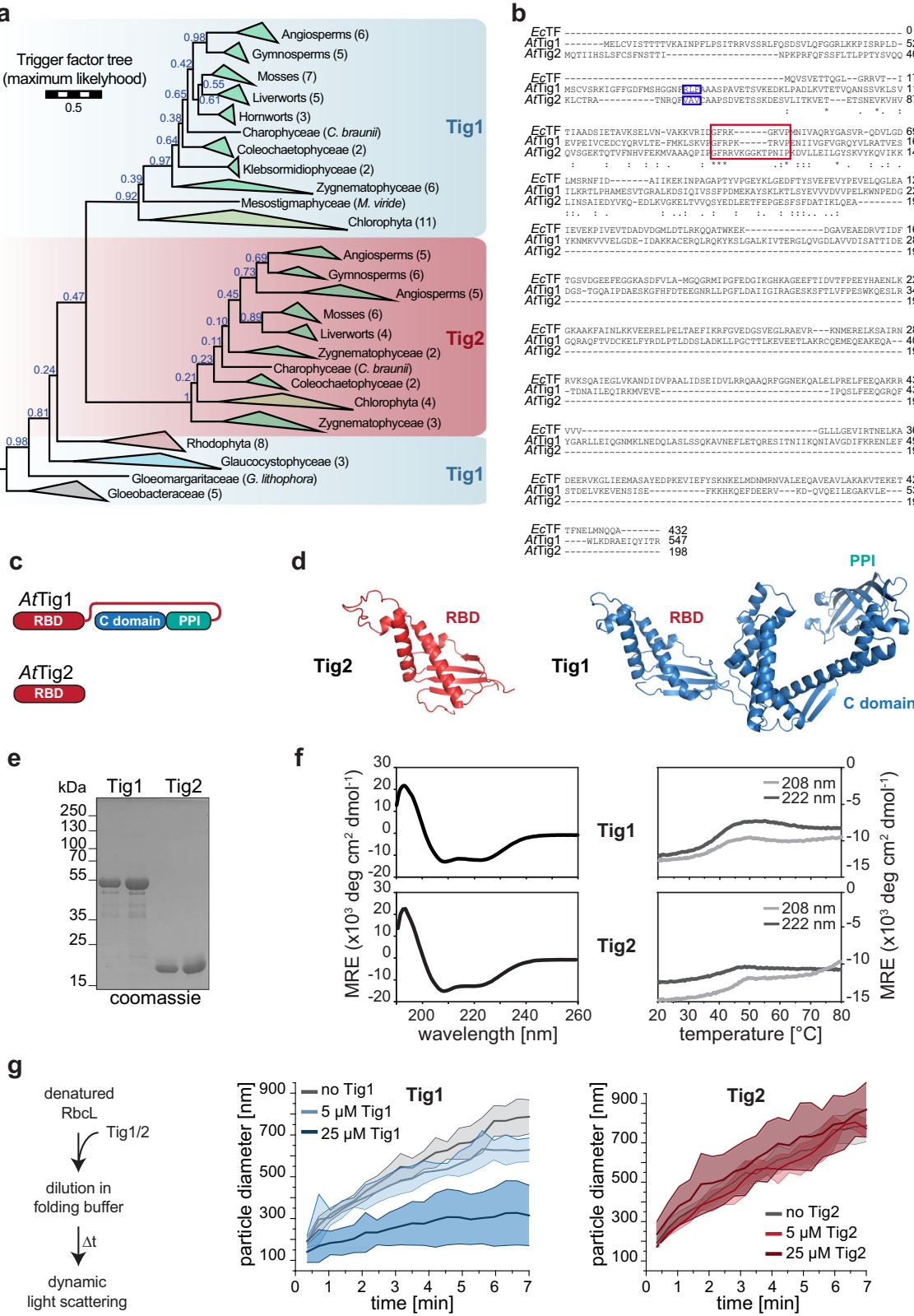

fusion under the 35S CaMV promoter, respectively. When grown under standard conditions, the Tig1-GFP and Tig2-GFP lines exhibited a phenotype that was indistinguishable from the Col-0 control. Laser-scanning confocal microscopy revealed an even distribution of Tig1-GFP within chloroplasts and an overlap with chlorophyll fluorescence (Fig. 2a). Similarly, the Tig2-GFP signal was also found throughout chloroplasts (Fig. 2b). Since Tig2-GFP expression showed additional

spherical foci, we performed immunofluorescence microscopy with the Tig2 antibody on protoplasts. The absence of these foci in the immunofluorescence images suggests that the observed foci may result from Tig2-GFP overaccumulation in plastids (Supplementary Fig. 4).

Tissue-specific gene expression analyses[22,23] showed highest transcript levels of *TIG1* and *TIG2* in the leaves of the vegetative rosette

**Fig. 1 | Two isoforms of chloroplast trigger factor are found in land plants.**
**a** Phylogeny of Trigger factor 1 and 2 sequences. Blue numbers give non-parametric (Felsenstein) bootstraps, numbers in brackets denote number of taxons per clade. The scale bar indicates how many amino acid substitutions happen on average per site along the horizontal branches (full phylogeny in Supplementary Fig. 1).
**b** Sequence alignment of the two chloroplast trigger factor homologs Tig1 (*At*Tig1, At5g55220) and Tig2 (*At*Tig2, At2g30695) and the ortholog of *E. coli* (*Ec*TF). Boxes indicate the predicted cleavage site of the chloroplast transit peptide (blue) and the putative ribosome binding motif (red). Asterisks mark perfectly conserved amino acids, similar residues are marked by dots. **c** Domain arrangement of *At*Tig1 (the N-terminal ribosome binding domain, RBD, in red; the PPIase domain in turquoise and the C-terminal chaperone domain in blue) and *At*Tig2. **d** Models of predicted conformation of Tig1 and Tig2 based on AlphaFold[16,17]. **e** Coomassie-stained SDS page of 1 and 2 μg purified mature Tig1 and Tig2 protein (lacking the N-terminal

chloroplast transit peptide), respectively. **f** Secondary structure content of Tig1 and Tig2, determined by Circular Dichroism (CD). Left panel: CD spectra of 0.1 mg/mL purified Tig1 protein (top) or Tig2 protein (bottom) measured at 20 °C. Right panel: Thermal unfolding is shown for the two minima at 208 nm and 222 nm during gradual heating from 20 °C to 94 °C. MRE mean residue ellipticity. **g** Chaperone activity assay of Tig1 and Tig2. Left panel: experimental setup. Chemically denatured RbcL protein was diluted to 0.5 μM in folding buffer in the absence or presence of 5 or 25 μM purified Tig1 or Tig2 protein, respectively. Dynamic light scattering (DLS) was monitored at 25 °C. Middle and right panel: Changes of hydrodynamic particle size (given as distribution widths of z-average diameters) in the absence or presence of Tig1 or Tig2, respectively. Each data series represents the arithmetic mean values of 3–4 technical replicates and 1–2 biological replicates, deviations are displayed as ribbon plots. Additional experiments are shown in Supplementary Fig. 3.

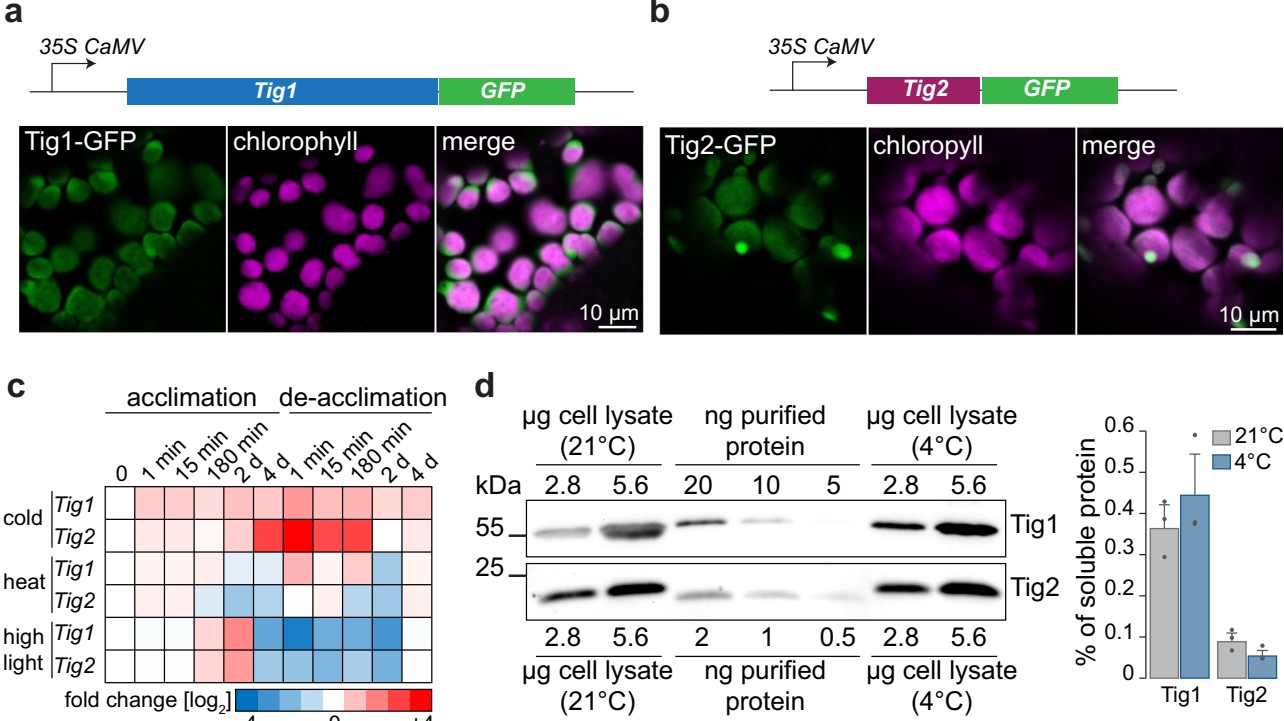

**Fig. 2 | Differential accumulation and chloroplast localization of Tig1 and Tig2.**
**a** Representative confocal microscopy images of Tig1 and **b** Tig2 accumulation in chloroplasts of Arabidopsis. GFP signal is shown in green, chlorophyll auto-fluorescence is shown in magenta. An additional overview of Tig2-GFP is presented in Supplementary Fig. 4. Experiments were performed in two biological replicates with two independent lines. **c** Heatmap illustrating transcript changes during heat (22 °C to 32 °C), cold (22 °C to 4 °C) and highlight (80 to 450 μmol photons m⁻² s⁻¹)

acclimation and de-acclimation. Transcript values were extracted from[24] and are given as log₂ fold change of Fragment Per Kilobase Million (FPKM) relative to the initial time point. **d** Quantitative immunoblot of Tig1 and Tig2 from leaf tissue of Col-0 grown for 4 weeks at 21 °C or plants, which were exposed for 1 weak at 21 °C and 3 weeks at 4 °C. Data are means from three biological replicates, recombinant protein served as standard, error bars represent standard deviations (SD).

(Supplementary Fig. 5). We further mined a system-level acclimation transcript dataset of Arabidopsis for putative altered *TIG1* and *TIG2* transcript accumulation upon environmental changes[24]. Interestingly, *TIG2* transcripts were strongly upregulated during cold acclimation, with an increase of up to 16-fold after 4 days of exposure to 4 °C, which was rapidly reduced to levels of standard conditions when the plants were transferred back to room temperature (Fig. 2c). Such strong increase was not observed for *TIG1*, although *TIG1* expression slightly increased during cold exposure (i.e., <threefold relative to room temperature conditions). The expression of *TIG1* and *TIG2* was not upregulated during heat exposure (32 °C) but showed a three to fourfold accumulation, respectively, after 2 days of high-light exposure (transfer from 80 to 450 μmol photons m⁻² s⁻¹) and a strong reduction during the respective acclimation phase (Fig. 2c). Remarkably, protein levels did not strictly follow their respective transcript levels. At room

temperature, protein concentration was 0.36 ± 0.06% for Tig1 and 0.09 ± 0.02% for Tig2, relative to all soluble proteins (Fig. 2d). Upon 21 days at 4 °C (with appearance of the mutant phenotype, below), Tig1 levels increased to 0.44 ± 0.1%, whereas Tig2 remained constant or rather decreased to 0.05 ± 0.02% (Fig. 2d).

## Deletion of Tig2 has a pronounced phenotype under cold conditions

To compare the function of the full-length trigger factor versus the truncated version, phenotypes of Tig1 and Tig2 loss-of-function mutants were characterized. The *tig1* T-DNA insertion line (SALK037730) has previously been characterized and carries a T-DNA insertion within the last exon[12]. The *tig2* insertion mutant (SALK110999) was determined to be homozygous based on PCR and sequencing, with the T-DNA occurring within the first intron (Fig. 3a and Supplementary Fig. 6).

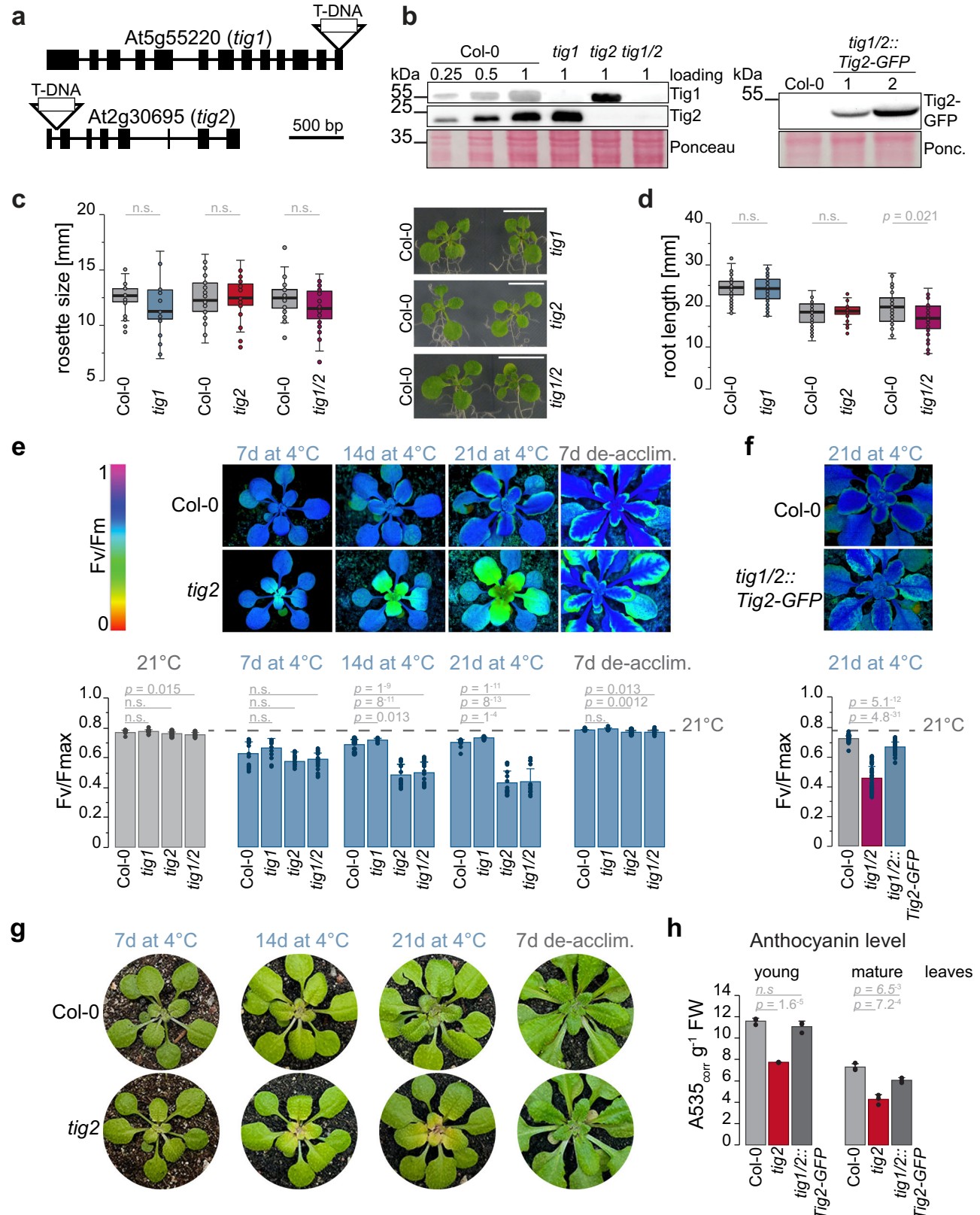

Furthermore, a homozygous *tig1*/*tig2* double mutant was established by crossing the individual lines (Supplementary Fig. 6) and absence of both proteins confirmed by immunoblotting (Fig. 3b). All lines displayed no obvious changes in leaf morphology under standard conditions. However, rosette diameter of double mutant seedlings was slightly, but not significantly, reduced when compared to Col-0 (Fig. 3c). The marginal

impact on plant growth of the double mutant was more evident by the reduced root length in 7-day-old seedlings grown on MS plates (Fig. 3d and Supplementary Fig. 7).

With the remarkable *Tig2* gene expression changes under chilling conditions, we expected a possible phenotype in the cold. A transition to 4 °C resulted in a general reduction in the maximum quantum yield

**Fig. 3 | Phenotype of trigger factor mutants.** Characterization of Arabidopsis trigger factor mutants. **a** T-DNA insertion site within the *tig1* gene at the 13th exon (line SALK037730) and within the first intron of *tig2* (line SALK110999). **b** Immunoblot of cell lysate of Co-0, *tig1*, *tig2*, and *tig1/2* lines (left panel) and of *tig1/2* lines complemented with *Tig2-GFP* (right panel). **c** Rosette size of Arabidopsis wild Col-0 and mutant seedlings (14 DAG) on MS- agar plates. Scale bar indicates 10 mm. **d** Root length of Arabidopsis Col-0 and mutant lines on MS- agar plates, grown vertically for 7 days after germination. Root length of seedlings grown on MS+ plates and images of the roots are shown in Supplementary Fig. 7. **c, d** Mutant are compared with the respective wild type, which were grown on the same plate. Boxplots shows data of >25 individual plants for rosette measurement and >36 individual plants for root measurement, grown in three independent biological replicates, respectively. Median is shown as solid line, box indicates lower and upper quartile, and the whiskers represent the data points that fall within 1.5 times the interquartile range (IQR) from the lower and upper quartiles. Any data point outside this range is considered as outlier. All samples are normally distributed after Kolmogorov-Smirnov, significant changes are indicated by unpaired two-tailed Student's *t*-test *p*-values; "n.s." not significant. **e** Top panel: False-color images of Fv/Fm chlorophyll fluorescence. Lower panel: maximum quantum yield of fluorescence (Fv/Fmax) representing photosynthetic activity of photosystem II. Fv/Fmax plots represent mean values of 3 biological replicates (with 5–10 technical replicates), error bars denote SD, two-sided Student's *t*-test *p*-values are given. **f** Photosynthetic activity of complemented line. Column plots show mean value and SD of 3 biological replicates (with 5–10 technical replicates). Two-sided Student's *t*-test *p*-values are given. **g** Images of young leaves of measured plants from (**e**). **h** Anthocyanin levels of young and mature leaves of 3-weeks cold-exposed plants. Mean values and SD of three biological replicates are plotted and two-sided Student's *t*-test *p*-values are given.

of PSII (Fv/Fm) in all lines, although this was recovered in Col-0 and the *tig1* mutant. In contrast, Fv/Fm levels remained approximately half in *tig2* and *tig1/2* compared to Col-0 levels (Fig. 3e). Following a return to 21 °C, photosynthetic defects were restored within seven days of de-acclimation, indicating that the absence of *Tig2* does not irreversibly damage leaf development. Importantly, *Tig2-GFP* expression in *tig1/tig2* lines almost completely complemented the observed phenotype, which validates that the absence of Tig2 is responsible for the observed defect (Fig. 3f). The reduced photosynthesis occurs in the chlorotic young leaves (Fig. 3g), which are particularly evident upon long-term cold exposure (Supplementary Fig. 8). Furthermore, lines lacking *tig2* exhibited markedly reduced anthocyanin levels during cold exposure (Fig. 3h), suggesting that *tig2* lines fail to upregulate these antioxidants to protect the plants during the acclimation to cold temperatures[25]. Expression of Tig2-GFP restored near normal levels of anthocyanin in both young and mature leaves.

To get a deeper understanding of the cellular defects associated with *tig2*, we compared the morphology of young leaves (from the inner rosette) between plants grown at 21 °C or at 4 °C. Transmission electron microscopy of 60 nm ultra-thin leaf cross-section showed only minor differences between the young leaves of Col-0 and the trigger factor mutants under standard conditions (Supplementary Fig. 9). *Tig1* mutants had more starch granules (Supplementary Fig. 10a), which is consistent with the previously observed altered energy balance of *tig1* mutants[12]. In the cold, starch content is also increased in *tig1* leaves and vacuoles are over-proportionally swollen in contrast to Col-0 and the *tig2* lines. Generally, all mutant leaf tissues have less dense spongy mesophyll in comparison to Col-0 leaves in the cold (Supplementary Fig. 9).

Observing the same samples at higher magnification indicated that there were no overall defects of chloroplast ultrastructure of the mutants at 22 °C and 4 °C. The thylakoid structure of these young leaves appeared less organized when compared to mature leaves (Fig. 4a). *Tig1* mutants again had increased starch accumulation and chloroplast size was significantly extended longitudinally in these mutants (Fig. 4b). Chloroplasts of *tig2* and *tig1/2* plants (grown at 22 °C) contained significantly fewer thylakoid membranes than those of Col-0 (Fig. 4c). This suggests that the development of Tig2-depleted lines may be somewhat retarded even under standard conditions (Fig. 4a and Supplementary Fig. 9). At 4 °C, it appears that all chloroplasts are swollen and more spherical, with rather loosely organized thylakoid membranes and increased stroma lamellae length (Fig. 4a and Supplementary Fig. 10b). All mutant chloroplasts contained fewer, yet significantly thicker, grana stacks (Fig. 4d), which may serve to increase maximal light use under low light[26]. It is possible that mutants adjusted their grana architecture to compensate reduced photosynthetic efficiency in the absence of Tig2 under cold conditions.

### Deletion of Tig2 causes impaired plastid ribosome biogenesis
We next performed shot-gun proteomics of densely grown seedling tissue, to reveal the molecular changes in the *tig2* lines. Two weeks after germination, Col-0 and *tig2* lines were either kept for additional 2 weeks at 21 °C or were subjected to a 3-week exposure at 4 °C (Fig. 5a). The longer growth for cold-treated plants was selected to have plants at comparable developmental stages. Mass spectrometric analyses of total lysates allowed to quantitatively compare 3043 proteins that were present in at least three out of four independent biological replicates and 3508 proteins at 4 °C (Supplementary Data 1). Replicates were highly reproducible with *Pearson's* correlation coefficients >0.95 within the respective Arabidopsis lines and conditions (Supplementary Fig. 11). Furthermore, the LFQ intensities were generally comparable between the Col-0 and the *tig2* lines from the respective condition ($r = 0.98$ at 21 °C and 4 °C, Fig. 5b), indicating that *tig2* mutants were not generally impaired on a proteomic level, which would have hampered a direct comparison.

Proteins with significantly altered LFQs between lines and conditions were extracted by two-sided *t*-test with a correction for multiple testing by permutation-based false discovery rate (FDR = 0.05, $S_0 = 1$), comparable to previous analyses[27,28]. Growth at 4 °C relative to 21 °C resulted in extensive changes of the plant cellular proteome with an increase in abundance for 26% of all determined proteins (in Col-0 and the *tig2* mutant) and a 32% or 29% reduction of all proteins within Col-0 and the *tig2* mutant, respectively. More than two-thirds of the up- or downregulated proteins showed similar trends in Col-0 and *tig2* (Fig. 5c). Consistent with the transcript data (Fig. 2c), Tig2 was significantly enriched during cold conditions ($\log_2$ of 0.86). The rather weak increase in the cold might be the reason why it appeared unchanged by immunoblotting (Fig. 2d). The volcano plots demonstrate that the Col-0 and *tig2* lines differed under ambient conditions by 19 and 4 proteins, respectively, which were significantly upregulated or downregulated in the mutant (Fig. 5d). Interestingly, the DEAD-box protein RH39, which supports formation of the characteristic 23S rRNA hidden break during plastid ribosome maturation[29] was the protein with the strongest upregulation (ninefold) in the mutant. The other prominently upregulated protein (eightfold) was Ntt2, the ATP/ADP nucleotide antiporter 2 protein of the inner envelope[30] (Fig. 5d).

During cold exposure, 88 proteins were upregulated and 45 were significantly downregulated in *tig2*, with the 23S rRNA processing factor HPE1/RBD1[31] being more than sevenfold upregulated (Fig. 5e). Among the significantly altered proteins, Gene Ontology categories[32] were only enriched for the proteins with reduced abundance in the cold (Fig. 5f). Most of these pathways are of chloroplast localization (chloroplast stroma: GO:0009570; chloroplast envelope: GO:0009941; thylakoid: GO:0009579), in line with the plastid-specific defects in the *tig2* mutant. Consistent with the reduction of anthocyanin levels in the *tig2* mutants (Fig. 3h), flavonoid synthesis processes are reduced in the mutants (GO:0009813). Chloroplast translation was among the processes with differential protein accumulation in *tig2* in the cold, with the 50S being the most enriched cellular compound in the samples of cold conditions (Fig. 5f). In fact,

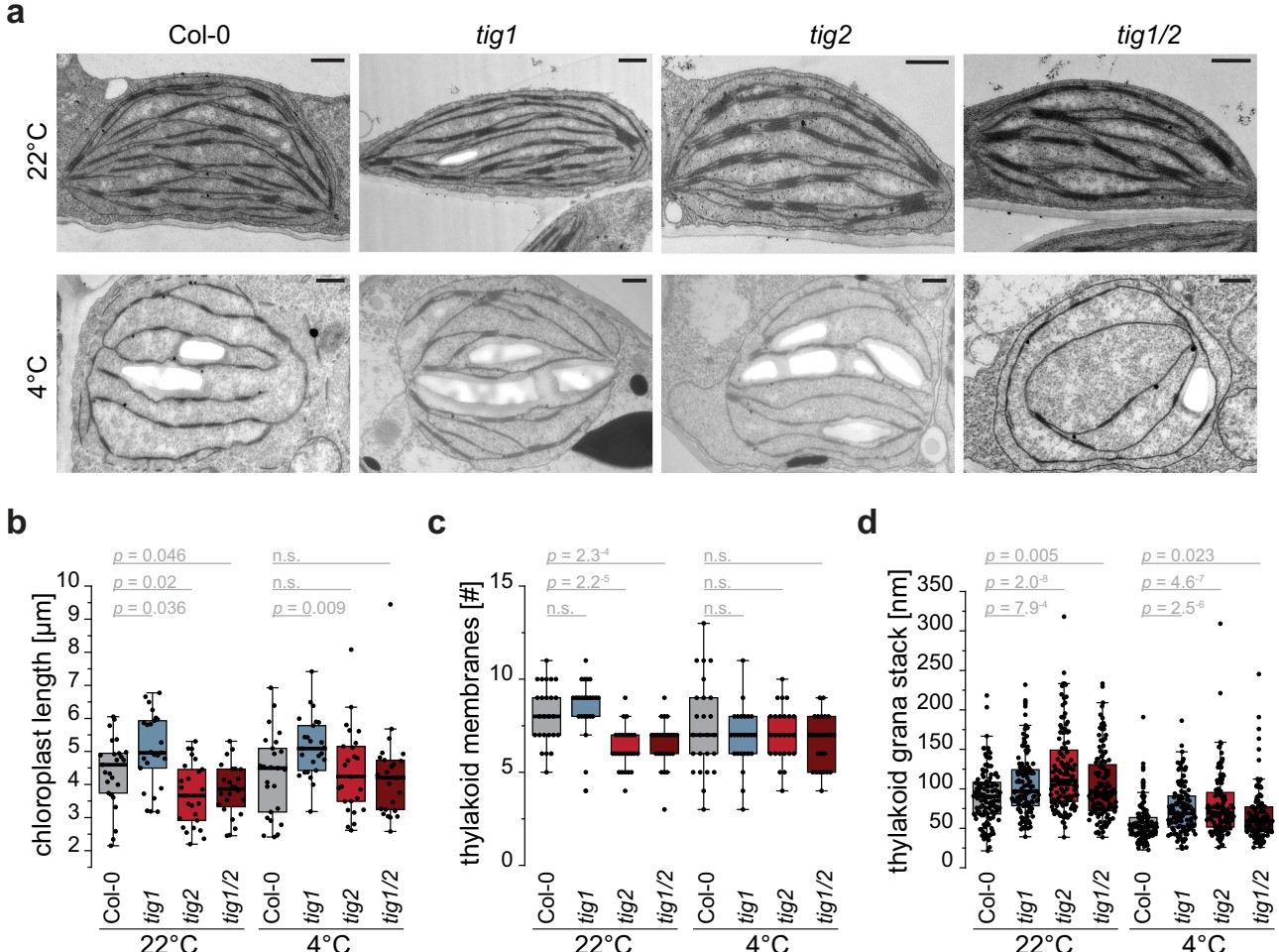

**Fig. 4 | Tig2 deletions induces altered chloroplast ultrastructures.** Electron microscopy derived ultrathin sections of chloroplasts from young leaves of Col-0 and trigger factor mutant lines. **a** Representative image of chloroplast ultrastructure of young leaves from five weeks-old plants that were kept at room temperature (22 °C) or transferred to 4 °C for the last two weeks before harvesting. Scale bar indicates 500 nm. **b** Chloroplast length, derived from random quantification of 25 chloroplasts per Arabidopsis line and condition, respectively. **c** Number of thylakoid membranes of 25 randomly quantified chloroplasts per line and condition, respectively. **d** Thickness of chloroplast grana stacks determined for 5 grana in 25 chloroplasts per Arabidopsis line and condition, respectively. All boxplots show median as solid line, box defining lower and upper quartile and whiskers that represent the data points that fall within 1.5 times the interquartile range (IQR) from the lower and upper quartiles. Any data point outside this range is considered as outlier. *P*-values of unpaired two-tailed Student's *t*-test are given. "n.s." not significant. Cross section of leaves and quantification of starch grana and stroma lamellae length are given in Supplementary Figs. 9 and 10, respectively.

most ribosomal proteins of the 50S showed reduced LFQ values in the cold-treated *tig2* samples, of which nine proteins were significantly reduced. These differences were considerably less pronounced at 21 °C. A similar, albeit less severe trend was observed for ribosomal proteins of the small subunit. This suggests that the absence of Tig2 affects ribosome accumulation particularly in the cold (Fig. 5g). In addition, protein levels of known plastidic DEAD-box RNA helicases and ribosome biogenesis factors were higher during cold exposure (Fig. 5h, left panel). A direct comparison between Col-0 and *tig2* revealed significantly increased accumulation of the helicases RH3, RH22, RH39 and RH50 and the ribosome biogenesis factors BPG2, RbgA, HPE1/RBD1 RNC3 (involved at various stages of rRNA maturation), and MORF2 (ribosomal protein transcript processing) under cold conditions. Furthermore, mTERF9 (ribosome assembly) and DG238/Iojap-related (potentially involved in ribosome activity) were only detectable in the *tig2* mutant in the cold, suggesting that plastid ribosome biogenesis might be specifically affected in the *tig2* mutant. Surprisingly, components of the photosynthesis machinery were not among the protein subgroup with pronounced changes (only Ycf3, PsaA, Elip1 and RbcL showed significant changes), suggesting that the lower photosynthetic activity is not the consequence of a general

defect in photosynthesis complex accumulation in the mutants (Supplementary Data 1). These changes collectively indicate that the observed changes result from defects in ribosome biogenesis rather than impaired photosynthesis.

Immunoblotting of samples from cold-treated young Arabidopsis leaves (based on same fresh weight) confirmed the specific defects within Tig2-deleted lines with reduced plastidic ribosomal proteins (e.g., uS12c, Fig. 6a). The Tig2 deletion affected the PSI and PSII core subunits PsaA and PsbA, respectively, without changing the accumulation of the inner PSII antenna CP43. In addition, RbcL was significantly reduced in these mutants, whereas *tig1* lines did not exhibit these defects (Fig. 6a). Deletion of both Tig species resulted in the lowest accumulation of RbcL, which agrees with the profound reduction of starch granules in the double mutant (Supplementary Fig. 10).

We next monitored, if chloroplast translation is defective in *tig2* by growing Col-0, *tig1*, *tig2* and *tig1/2* on plates with sublethal concentrations of 70S translation inhibitors. Even during growth at ambient conditions, *tig2* and *tig1/2* were hypersensitive to 20 μM lincomycin, and 40 μM chloramphenicol, respectively, whereas the absence of Tig1 did not affect growth (Fig. 6b and Supplementary

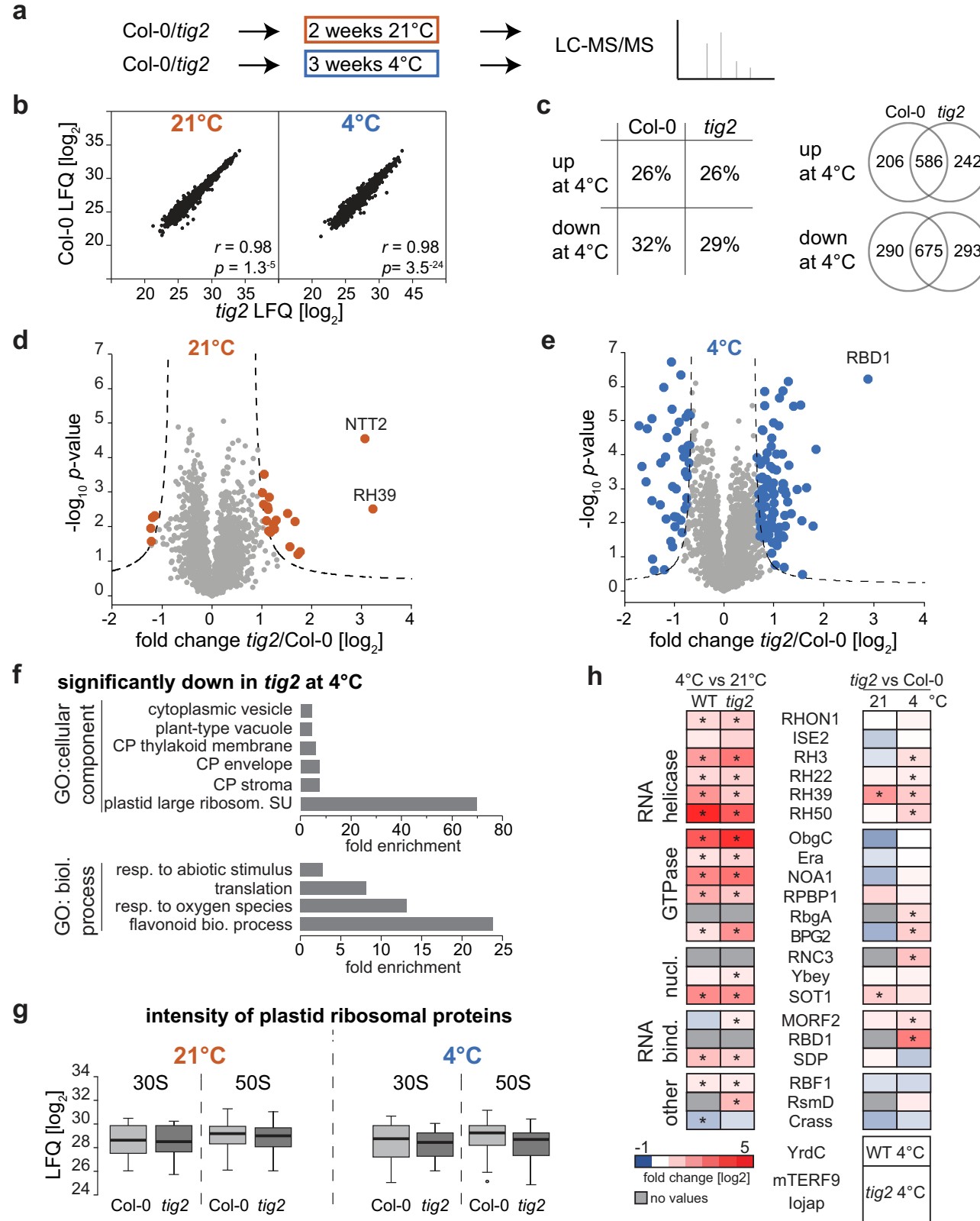

**f** **significantly down in *tig2* at 4°C**

**g** **intensity of plastid ribosomal proteins**

**h**

Fig. 12). Consequently, Tig2 appears to exert a direct influence on the fidelity of the chloroplast ribosome pool. The precise ribosome maturation defects within *tig2* mutants were examined by northern blot analyses, with probes directed against the final maturation products of the 23S (SSU) and 16S (LSU) subunits. Processing of the 23S rRNA fragments appeared to be specifically impaired under both growth temperatures. The unprocessed precursor of 2.9 kb and 2.4 kb accumulated in the mutant. In contrast, the 1.8, 1.3, and 1.1 kb fragments were markedly reduced. The hidden-break product of 0.5 kb was even below detection limits in the *tig2* samples (Fig. 6c and Supplementary Fig. 13). Interestingly, the 16S rRNA fragments were only reduced in the cold-treated *tig2* samples. This demonstrates that Tig2 facilitates biogenesis of the 50S subunit, which is particularly pronounced under cold conditions (Fig. 6c).

**Fig. 5 | Proteome differences reveal a role of Tig2 in chloroplast ribosome biogenesis.** Proteomic comparison between Col-0 and *tig2* mutant lines.
**a** Experimental setup: Plants were kept for two weeks at 21 °C or transferred to 4 °C for cold acclimation over three weeks. Whole seedlings were harvested and subjected to mass spectrometric analysis. **b** Direct comparison of protein abundance (given as mean value of log2-transformed LFQ, derived of four biological replicates) between Col-0 and *tig2*. *Pearson's r* and *ANOVA p*-values are given in the graph. **c** Left panel: percentage of significantly changed proteins (FDR 0.05 and $S_0 = 1$) between plants grown a room temperature or 4 °C; right panel: number of proteins that are significantly up- or downregulated in the two lines. **d**, **e** Volcano plots of the fold-changes plotted against the *p*-values of Col-0 and the *tig2* mutant for plants grown at 21 °C or 4 °C, respectively. The *p*-values derive from two-sided *t*-test, a minimal fold change $S_0 = 1$, and a permutation-based FDR < 0.05, with at least three valid values in both groups. **f** Gene Ontology (GO) enrichment for proteins with significant reduction in *tig2* compared to Col-0. For Biological Process, GO IDs are:

GO:0009813 for "flavonoid biosynthetic process", GO:0000302 for "response to reactive oxygen species", GO:0006412 for "translation" and GO:0009628 for "response to abiotic stimulus". For Cellular Component, GO IDs are: GO:0000311 for "large ribosomal subunit", GO:0009941 for "chloroplast stroma chloroplast envelope", GO:0009535 for "chloroplast thylakoid membrane", GO:0000325 for "plant-type vacuole" and GO:0031410 "cytoplasmic vesicle". **g** Box blots of $\log_2$ LFQ values of chloroplast ribosomal proteins of the four biological replicates with median shown as solid line, box indicating lower and upper quartile and the whiskers defining data points that fall within 1.5 times the interquartile range (IQR) from the lower and upper quartiles. Data point outside this range is considered as outlier. **h** Heat map visualizing $\log_2$ fold changes between 21 °C and 4 °C grown Col-0 and *tig2* lines (left panel) and changes between Col-0 and *tig2* (right panel). Proteins which were only detected in one line are shown below the graph. Asterisks indicate proteins with significant changes (two-sided *t*-test, a minimal fold change $S_0 = 1$, and a permutation-based FDR < 0.05) for the respective condition.

With the existence of a ribosome binding domain (Fig. 1), Tig2 should interact with plastidic 70S ribosomes. Indeed, our sucrose cushion assays of Col-0 samples from seedlings at 21 °C showed that Tig2 co-sediments with ribosomes. Interestingly, the fraction of ribosome-co-sedimented Tig2 was approximately threefold higher than that of Tig1 (-25% versus -8%, respectively). This implies that a substantial proportion of chloroplast Tig2 interacts with ribosomes (Fig. 7a). While Tig1 ribosome interaction was puromycin sensitive, Tig2-ribosome association was high-salt-sensitive but puromycin did not induce significant changes. The absence of Tig2 appears to significantly impact ribosome-association of Tig1, which could be the consequence of a reduced pool of translationally competent chloroplast ribosomes. To test, if Tig2 might indeed bind to the tunnel exit site, we exploited the most recent interaction predictions provided by the AlphaFold 3 server[33]. Providing the Arabidopsis 23S rRNA and the corresponding ribosomal proteins that surround the tunnel exit site (uL22c, uL23c, uL24c, uL29c, uL32c), yielded models that were highly comparable to the Cryo-EM structures of spinach chloroplast ribosomes (Fig. 7b)[34,35]. In these models, Tig2 was always predicted to bind at the tunnel exit, with its positively charged loop domain facing the tunnel (Fig. 7b). Closer analyses of the interaction site showed that Tig2 is predicted to form four contacts with the ribosomal protein uL24c. It its noteworthy that for all predictions done, Tig1 and Tig2 were never predicted to simultaneously bind the tunnel exit site. In these runs, only Tig2 was shown to interact with uL24c, suggesting that common Tig1 and Tig2 ribosome interaction is sterically not possible. Certainly, experimental data are needed to confirm these predictions, however, the regular prediction of the Tig2 contact near the tunnel exit, suggests that Tig2 may be indeed a ribosome associated factor. Additionally, we compared co-migration of Tig1 and Tig2 with polysomes in sucrose gradient fractions of Arabidopsis lysates. While Tig1 showed the expected migration pattern, deep into polysome fractions[12], we could not detect any Tig2 signal in polysome fractions. Instead, most signal derived from fractions with lower molecular weight (Supplementary Fig. 14a), suggesting that the Tig2 ribosome contact might occur on ribosome biogenesis intermediates or free 50S particles, again supporting the hypothesis that Tig2 functions during ribosome biogenesis. We then performed mass spectrometric measurements of ribosomal particles from the co-sedimentation assay comparing Col-0 and *tig2* and growth at 21 °C or 4 °C. A total of 1046 and 790 proteins could be quantified from the pellet of ambient and cold-acclimation samples, respectively (Supplementary Data 1). In seedlings grown at 21 °C, only a few plastidic ribosomal proteins were reduced in the ribosomal pellet of *tig2* lines. However, cold treatment resulted in significant changes of 11 ribosomal proteins of the SSU and 9 proteins of the LSU (Supplementary Fig. 14b). Of note, not all ribosomal proteins were reduced in the pellet of *tig2* samples, which provides further evidence that the ribosome biogenesis trajectory is disrupted in the cold, when Tig2 is absent. Consistent with the

aforementioned findings, several plastidic ribosome biogenesis factors showed altered ribosome association in the *tig2* mutant. Interestingly, RH39 a helicase that is postulated to facilitate the formation of the hidden break in 23S rRNA was found to be enriched (>sevenfold at 21 °C and >twofold at 4 °C) in ribosomal fractions of the *tig2* samples (Fig. 7c and Supplementary Data 1). This supports the hypothesis that maturation of the 50S subunit is disrupted in these mutants. However, the proteomics data indicated no significant alterations in the ribosome-association of other co-translationally acting nascent chain biogenesis factors[10,28] in the *tig2* mutant, including the chaperonin machinery, Hsp70, Tig1 and cpSRP54 (Fig. 7d). Consequently, Tig2 does not appear to be specifically involved in recruiting these factors to translating ribosomes.

Taken together, the deletion of the truncated Tig2 protein results in impaired chloroplast ribosome biogenesis, which is particularly pronounced under cold conditions. The defect in 50S maturation is most evident by impaired post-maturation cleavage of the 23S, which is essential for the proper functioning of the plastidic protein synthesis machinery.

## Discussion

Maturation of chloroplast-encoded proteins starts on the level of translation and Tig1 is presumably the first chaperone in this cascade[10,12]. The deletion of Tig1 has no severe phenotype in algae and land plants[12]. However, several data indicate that chloroplast homeostasis was disturbed in the absence of Tig1. The deletion mutants displayed an elevated energy demand in Chlamydomonas and Arabidopsis, with increased photosynthetic cyclic electron flow, presumably to ensure energy supply[12]. The higher number of starch granules of the Arabidopsis *tig1* mutant observed here (Fig. 4), agrees with these previous observations. The increased energy demand of the *tig1* mutants could derive from protein misfolding of nascent polypeptides, which is compensated by increased activity of other chaperones or increased translation. Such an energy demand might be only tolerated under optimal laboratory conditions but would have drastic consequences in the natural environment.

The missing growth defect of *tig1* mutants could be due to a compensatory role of Tig2, at least in land plants (Fig. 1a). However, our data indicate that a backup role for Tig2 is unlikely: i. simultaneous deletion of Tig1 and Tig2 did not result in a growth defect under standard growth conditions (Fig. 3c); ii. For our two model substrates tested, Tig2 exhibited no aggregate preventing activity, indicating that Tig2 may not bind unfolded proteins as observed for Tig1 (Fig. 1g); and iii. deletion of Tig2 has a specific phenotype under cold conditions, which was not observed in the *tig1* deletion background (Figs. 3 and 4). Yet, Tig2 associates with plastid ribosomes, at a higher rate than Tig1 (Fig. 7a). Our proteomic screen of the *tig2* mutants, grown at 21 °C or 4 °C revealed the role of Tig2 during ribosome biogenesis (Fig. 5). In the *tig2* mutants, the energy regulatory protein NTT2 and RH39 were

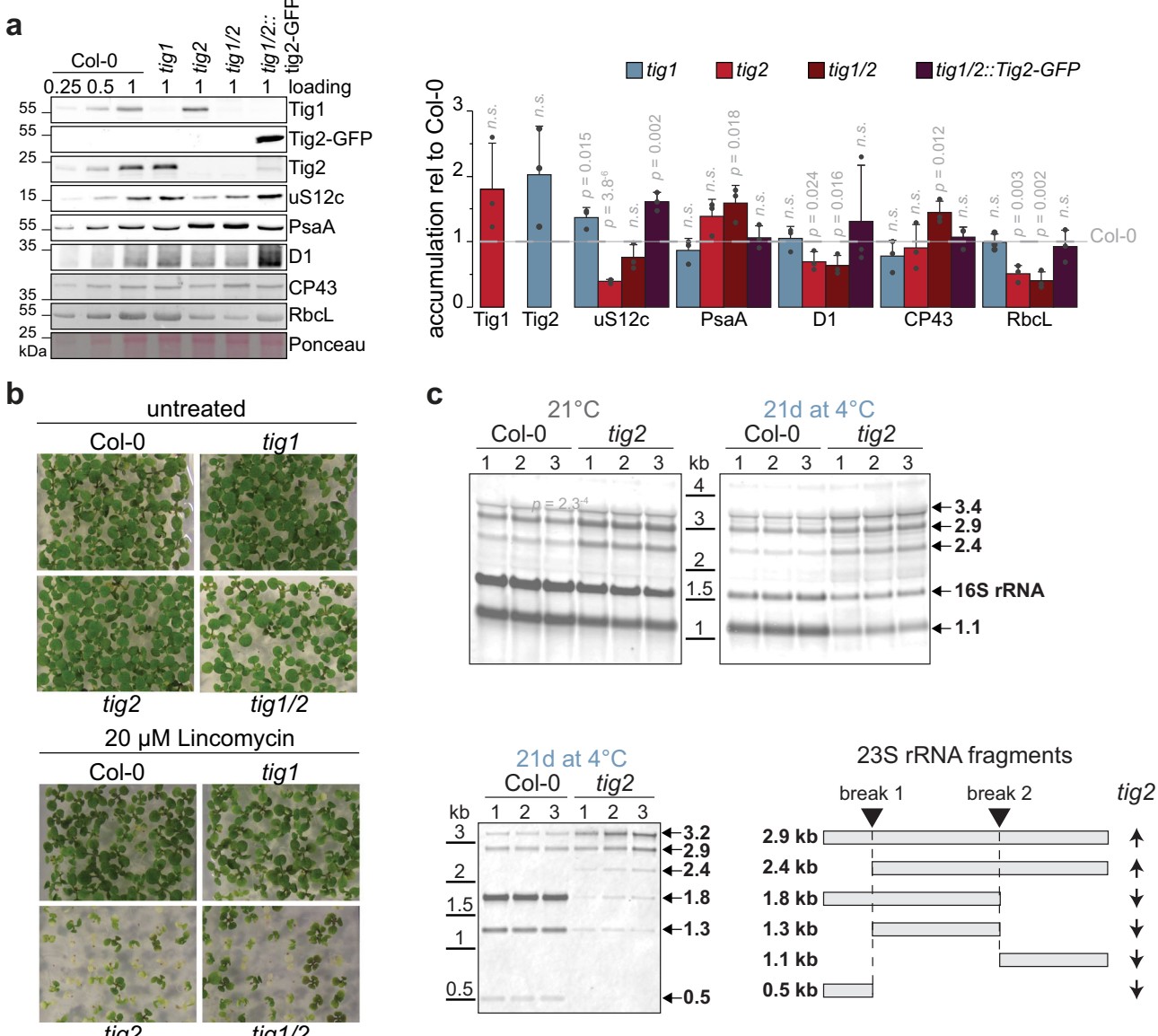

**Fig. 6 | Ribosome biogenesis defects caused by Tig2 deletion. a** Left: Immuno-blot monitoring accumulation of selected chloroplast proteins in Col-0 and trigger factor mutant lines. Protein samples were collected from young leaves after 21 days of cold-exposure; right panel: quantification of bands. Mean values derive from three biological replicates, error bars denote SD, two-sided Student's *t*-test *p*-values are given, "n.s." not significant. **b** Seedlings were grown for 10 days (DAG) on plates with and without lincomycin. **c** RNA was isolated from young leaves of Col-0 and *tig2* lines after 2 weeks of cold exposure or permanent growth at 21 °C. and analyzed by Northern blot (1 µg per lane). Probes against chloroplast 16S and the possible post-maturation fragments of chloroplast 23S (indicated by arrows and depicted in the lower right panel). Quality of RNA and loading controls and are shown in Supplementary Fig. 13.

the most significantly upregulated proteins under ambient conditions. RH39 is a DEAD-box helicase that introduces so-called hidden-breaks in the 23S rRNA. The absence of RH39 results in aberrant translation in the chloroplast[29]. Interestingly, the Arabidopsis *RH39* mutant (*NARA12*), displayed similar features like the *tig2* mutant (reduced photosynthesis and lower levels of RbcL), yet growth retardation is more pronounced in the absence of RH39[29]. In the cold, profoundly more plastid ribosome biogenesis factors accumulated and ribosomal proteins of the 50S subunit were reduced in *tig2* (Fig. 5g, h). Translation and ribosome assembly have recently been identified as the super-hubs of the plant's acclimation program to heat, high light and cold in a multi-omics analysis[24]. This is consistent with previous reports of cold defects that become evident if factors involved in chloroplast ribosome maturation are deleted (e.g., Hcf7, DIM1/PFC1/Rpl33, Svr3, CP29A, CP31A, RH3, CGL20, CGL38 or RH50) (reviewed in ref. 8).

Importantly, defects of ribosome biogenesis can also result in cold sensitivity in bacteria (reviewed in ref. 7). Thus, the biogenesis and assembly of ribosomes appear to be particularly susceptible to cold exposure, necessitating the involvement of specific auxiliary factors.

It has been demonstrated that molecular chaperones contribute to ribosome biogenesis in *E. coli* particularly during exposure to cold or heat (reviewed in ref. 7). For example, ribosome assembly is impeded at elevated temperatures when the Hsp70/DnaK and its co-chaperones are lacking[36]. Additionally, bacterial trigger factor is proposed to facilitate ribosome assembly, and the *tf* mutant displays cold sensitivity[13,37]. Yet, deletion of chloroplast Tig1 has no obvious consequence for ribosome biogenesis, at least judged by the absent phenotype during cold conditions (Figs. 3 and 4). Thus, the isolated ribosome binding domain of Tig2 seems to possess a distinctive function in this process. Northern blot analysis of the chloroplast rRNA

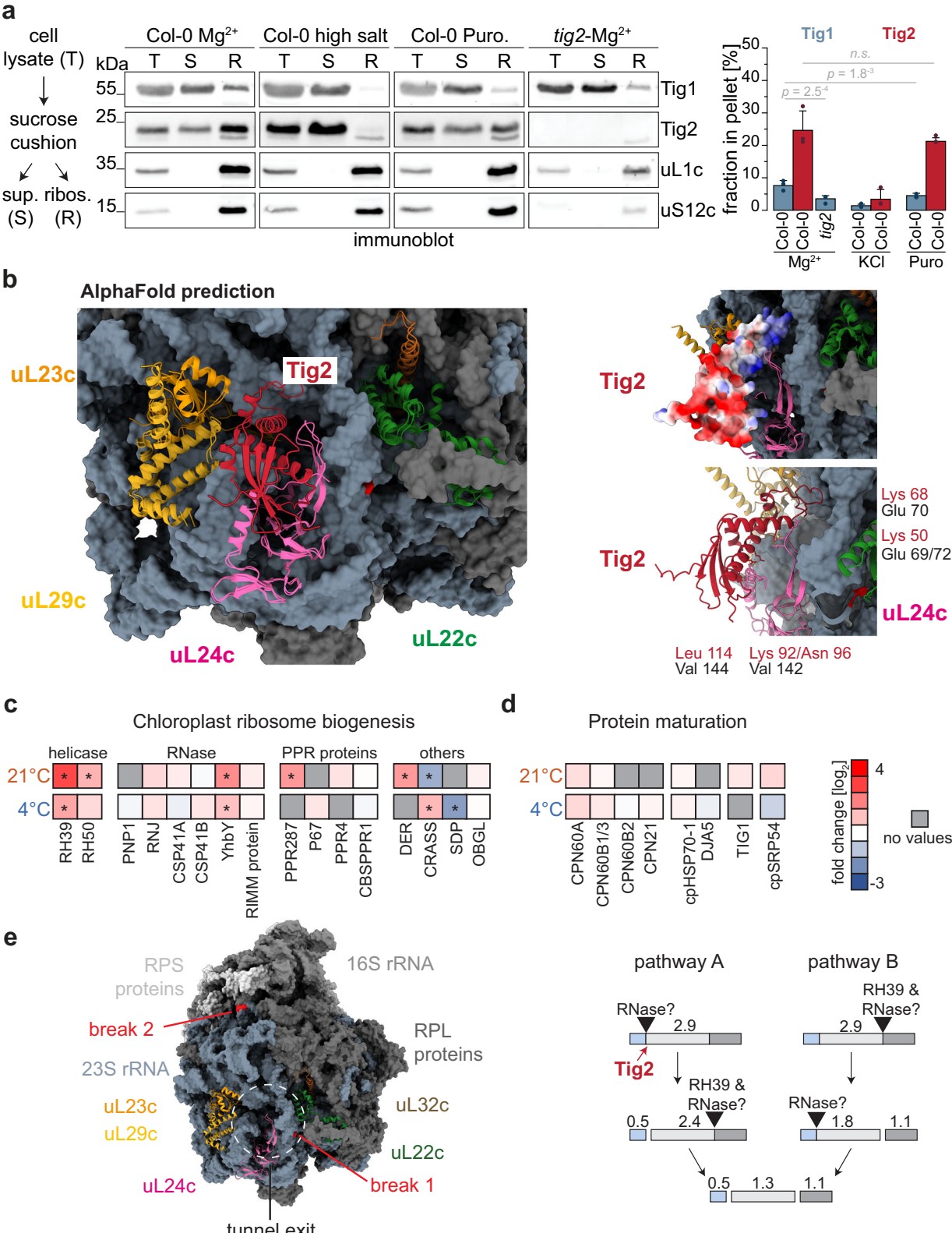

**a** Cell lysate (T) → sucrose cushion → sup. (S) ribos. (R); immunoblot for Tig1, Tig2, uL1c, uS12c across Col-0 Mg²⁺, Col-0 high salt, Col-0 Puro., and tig2-Mg²⁺. Bar graph: fraction in pellet [%] for Tig1 and Tig2.

**b** AlphaFold prediction showing Tig2 with uL23c, uL29c, uL24c, uL22c.

**c** Chloroplast ribosome biogenesis; **d** Protein maturation; **e** Ribosome structure and pathways A and B.

revealed that post-maturation cleavage of the 23S rRNA into the 0.5, 1.3, and 1.1 kb fragments is disrupted (Fig. 6c). The formation of these hidden breaks is thought to be mediated by the DEAD-box protein RH39 and undiscovered RNases. It is likely that RH39 is particularly important for hidden break 2 (Fig. 7e)[29], and a recently described Cgl20 protein may be needed for the introduction of break 1[29,38]. The specific processing defects of the *RH39, cgl20* and the *tig2* mutants are unique

in comparison to the general rRNA maturation defects determined in other ribosome biogenesis mutants. The Tig2 deletion causes strong reduction of the 0.5 kb fragment, which was unaffected in the *RH39* and *cgl20* lines, (Fig. 6c). The elevated accumulation of RH39 on the proteome level and in pelleted ribosomes (<sevenfold at 21 °C and >twofold at 4 °C; Figs. 5h and 7c) of *tig2* mutants provides further evidence for an impaired hidden break formation. Thus, Tig2 might be

**Fig. 7 | Action of ribosome-associated Tig2. a** Left: Schematic workflow of ribosome co-sedimentation assay. Immunoblot of total protein (T) as input control, supernatant fraction (S), and the pellet sample (R) containing the enriched ribosomes. Samples derive from 3-week-old densely pot-grown plants, treated with standard ribosome buffer (Mg²⁺), with 800 mM KCl (high salt) or 1 mM puromycin (Puro). Right: Quantification of immunoblot (R/(R + S)) from three independent experiments, mean values, SD and two-sided Student's *t*-test *p*-values are given. **b** Structural prediction of the Arabidopsis 23S rRNA, relevant ribosomal protein surrounding the tunnel exit site and Tig2, based on AlphaFold 3. Predicted Arabidopsis models are superimposed with the Cryo-EM 6ERI structure of spinach[34]. Postulated Tig2-ribosome interaction is shown in the right panels with Tig2 shown as surface charge (top) and ribbon model (bottom). Predicted contacts between amino acids of Tig2 and uL24c are shown. **c** and **d** Mass spectrometric quantification of proteins that were present in ribosomal pellets of the category chloroplast ribosome biogenesis (**b**) and co-translational protein maturation (**c**). Col-0 and *tig2 lines* were kept for 21 days in the cold (4 °C) or at standard conditions (21 °C) for two weeks. Heatmaps represent fold-change differences (log₂) between Col-0 and *tig2* lines. **e** Left: model of the chloroplast ribosome derived from the 6ERI Cryo-EM structure[34]. Relevant ribosomal proteins surrounding the tunnel exit site are marked in color; positions of 23S rRNA breaks are indicated in red. RPS ribosomal protein of the small subunit, RPL ribosomal protein of the large subunit. Right: Schematic overview of final rRNA maturation and the role of RH39 according to[29]. Postulated action of Tig2 is indicated.

directly involved in hidden break formation, however, rRNA binding activity of a trigger factor domain was not reported in literature, to date. There is an interesting analogy for the maturation of cytosolic 80S ribosome. 80S ribosome biogenesis depends on so-called place-holders, which temporarily bind to assembly precursors to shield off sensitive ribosome interaction surfaces from unwanted interactions (reviewed in ref. 39). Recently, the human ErbB3 receptor-binding protein (Ebp1) and the respective yeast homologue "60S pre-ribosomal nuclear export factor", Arx1 were postulated to exhibit such a role in shielding off the ribosomal exit tunnel site on 80S ribosomes[40,41]. The non-essential Ebp1 and Arx1 proteins appear to adopt the conformation similar to that of the co-translationally acting methionine aminopeptidase (MetAP), despite lacking a functional active site[42]. Ebp1 appears bound to translating ribosomes during the early phase of elongation and is subsequently outcompeted when other co-translational factors, including processing and sorting factors and molecular chaperones bind the emerging nascent polypeptide[40]. Ebp1 is abundantly associated with 80S ribosomes, at higher ratios compared to other co-translationally acting factors[40,43]. Also, Tig2 seems to co-sediment at higher levels with chloroplast ribosomes (3× compared to Tig1, Fig. 7a) and it is intriguing to speculate that Tig2 may present an analogous factor in chloroplasts. During biogenesis of bacterial 70S ribosomes, the solvent-exposed exit tunnel site forms at an early stage of 50S assembly[44]. In plastid ribosomes, this domain also includes rRNA helix H25, which contains hidden-break 1 (Fig. 7e). Correct folding of this section is crucial for the maturation of the 50S domain containing hidden break 2 in rRNA helix H63, the site which is introduced by help of RH39[29,35,44]. It could be envisioned that Tig2 evolved, in order to protect the tunnel exit site during formation. The absence of Tig2 may disturb ribosome assembly, particularly under the challenging cold conditions, which also affects the formation of the hidden break by RH39 and the accompanying RNases. Of note, such place holders are not reported for the biogenesis of the bacterial 70S ribosomes (reviewed in ref. 7). Consequently, Tig2 may represent a unique acquisition of a chaperone domain for the maturation of the plastidic 70S ribosomes. Curiously, the Arabidopsis Ebp1 homolog (At3g51800) is fourfold enriched in ribosomal pellets of *tig2* mutants versus Col-0 under cold conditions (Supplementary Data 1) and may point to cross-communication between chloroplast and cytosolic translation during challenged plastid ribosome biogenesis. It is furthermore interesting, that a small globular protein, termed mL105, interacts with bL23m of mitochondrial ribosomes in *Tetrahymena thermophila* and *Polytomella magna*[45,46]. mL105 has homology to the M-domain of prokaryotic-type signal factor Srp54 and was hypothesized to contribute to membrane targeting of mitochondrial ribosomes[45,46]. Alternatively, mL105 could represent another strategy how a truncated factor evolved to serve the correct maturation of the ribosomal exit tunnel interface. However, functional analysis of mL105 is needed to proof a comparable function like that of Tig2 and EBP1/Arx1.

Our proteomic comparison of ribosomal pellets between Col-0 and *tig2* also revealed several ribosomal proteins of the small subunits with significantly altered abundance in the *tig2* mutant in the cold (Supplementary Fig. 14b). Of the 6 proteins with reduced abundance, bS1c, bS2c and bS5c accumulated to lower levels in proteomic data of *tig2* lysates. We can only speculate at this point why loss of Tig2, a biogenesis factor of the large subunit, affects biogenesis of the 30S subunit. But it was reported that biogenesis of the subunits is interconnected and that maturation of the cytosolic 60S subunit affects maturation of the 40S particle (e.g.,[47]).

It remains to be demonstrated by structural analyses how ribosomes are impaired during maturation in the absence of Tig2 and how Tig2 performs its function. Furthermore, it would be interesting to understand why most chlorophytes do not require Tig2 during cold exposure, whereas it is of importance for most streptophytes.

## Methods
### Plant material
Unless otherwise specified, *Arabidopsis thaliana* plants were cultivated under short-day conditions (21 °C, 14 h light at 120 µmol of photons m⁻²s⁻¹, 10 h dark). For cold treatment, densely grown plants were transferred to 4 °C for 21 days (3 weeks after germination), under short-day conditions. Individual-pot-plants, were grown for 5 weeks after germination before transfer to cold. Seeds were obtained from the Nottingham Arabidopsis Stock Centre (*tig2* T-DNA insertion line SALK110999[48];). The *tig1* mutant has been previously described[12]. The double mutant with deletion of the *Tig1* and *Tig2* alleles was generated by crossing the respective mutants. To determine the root length and rosette diameter of the seedlings, plants were grown on ½ Murashige and Skoog (MS)+/− agar plates under long-day conditions (16 h day, 8 h dark). For root measurements, plates were grown vertically for 7 days.

### Sequence analyses
Trigger factor sequences were derived from http://www.uniprot.org and http://phytozome.jgi.doe.gov (for accession numbers see Table S1). Sequence alignment was performed with Clustal Omega (https://www.ebi.ac.uk/Tools/msa/clustalo). For the phylogenetic analysis, Tig1 and Tig2 protein sequences from Cyanobacteria, Glaucophyta, Rhodophyta, Chlorophyta and Streptophyta were collected from the NCBI and ONEKP databases using BLAST. Sequences were aligned with MAFFT 7.110 using the iterative refinement method with local pairwise alignment information (L-INS-i). Chloroplast transit peptide (cTP) sequences were predicted using TargetP 2.0 (https://services.healthtech.dtu.dk/services/TargetP-2.0/) and subsequently removed from the alignment. Lineage-specific insertions were manually removed from the alignment. A maximum likelihood phylogeny was inferred using IQ-TREE 2.0.3. ModelFinder was used to determine the best-fitting model according to the corrected Akaike information criterion (cAIC) as the general matrix (LG) with empirical amino acid frequencies and the FreeRate model with 5 categories to account for rate heterogeneity across sites (LG + F + R5). The resulting tree was rooted using sequences from basal cyanobacteria of the family Gloeobacterales as an outgroup. Standard non-parametric

(Felsenstein) bootstraps were calculated in 100 replicates using IQ-TREE 2.0.3 to assess the robustness of the phylogenetic inference.

## Anthocyanin measurement

Anthocyanin accumulation was determined from leaves 1–5 (young) and leaves 6–8 (mature), which were frozen in liquid nitrogen. After cryogenic grinding of the material, 50 mg fraction were extracted with 1 mL of extraction buffer ($H_2O$, 2-propanol and HCl 81:1:18). The samples were incubated for 3 min at 95 °C and 650 rpm and incubated in the dark overnight at room temperature. After centrifugation at $16,000 \times g$ for 15 min, the absorbance of the supernatant was measured at 535 nm and 650 nm. The amount of anthocyanin was calculated using the formula: $(E_{535} - 2.2 \times E_{650}) \times$ (g fresh weight)$^{-1}$.

## Protein purification and in vitro assays

Cloning of the construct for heterologous expression of Arabidopsis Tig1 was published before[11]. Chaperone activity assays via Dynamic light scattering were performed as described in ref. [12]. Circular Dichroism were measured as in ref. [11]. For Tig2 expression, the coding sequence was amplified from Arabidopsis cDNA with primers Tig2NdeI-F and Tig2EcoRI-R (Table S2) and cloned with NdeI and EcoRI into pTyb21 (NEB) giving construct pFW190. Tig2 was purified as described for Tig1[11].

## GFP-tagged constructs, confocal microscopy, and electron microscopy

CaMV 35S::Tig1-GFP (pGGZ_SK13) and CaMV 35S::Tig2-GFP (pGGZ_SK11) constructs were generated using the GreenGate cloning system[49]. Coding sequences were amplified from Arabidopsis Col-0 cDNA using primers Tig1-GFP-F & Tig1-GFP-R for the Tig1 construct and Tig2-GFP-F & Tig2-GFP-R for Tig2 (Table S2). For both constructs, the GFP module was fused at the 3´-end of the coding sequence. For selection, the seed coat marker pGGF Alli-mCherry was used as module instead of a resistance marker.

To generate stable Arabidopsis lines, both constructs were initially transformed into *Agrobacterium tumefaciens* (GV3101 containing pMP90). Subsequently, Col-0 and the *tig1/2* mutant line were transformed via a modified floral inoculation method[50]. To this end, 2 mL *A. tumefaciens* overnight culture, carrying the respective plasmids, were softly centrifuged and the cell pellets resuspended in 2 mL 5% (w/v) sucrose solution. After addition of 0.02% (v/v) Silwet L-77, 5 µL droplets were applied on all flowers of 4–8 individual plants per line. Plants were then temporarily covered and kept at reduced light conditions for 2 days. Transformed seeds were selected by detection of the mCherry seed coat marker and propagated until homozygous plants could be identified.

Confocal images were captured with a Zeiss LSM880, AxioObserver SP7 confocal laser-scanning microscope, equipped with a Zeiss C-Apochromat 40×/1.2 W AutoCorr M27 water-immersion objective. Fluorescence signals of GFP and chlorophyll autofluorescence (excitation/emission 488 nm/ 500–571 nm and 543/579-718), were processed using the ZEN blue or Fiji software.

For transmission electron microscopy, plants were grown in a growth cabinet at 22 °C under short-day conditions (10 h/14 h light/dark; 120 µmol m$^{-2}$ s$^{-1}$). After 3 weeks, the plants were transferred to 4 °C for cold treatment or kept at 22 °C as a control. Before fixation, the plants were stored in the dark for 14 h to prevent accumulation of starch. Young leaves were cut into 1 mm$^2$ pieces and stored in fixation buffer (75 mM sodium-cacodylate, 2 mM $MgCl_2$, pH 7 and 2.5% (v/v) glutaraldehyde) at 4 °C for at least 2 days. Fixation was carried out as described before[51]. After post-fixation with 1% (w/v) $OsO_4$, samples were contrasted *en bloc* with 1% (w/v) uranyl acetate in 20% acetone, dehydrated with a graded acetone series and embedded in Spurr's resin. Ultrathin sections of approximately 60 nm were contrasted with lead citrate and examined with a Zeiss EM 912 transmission electron microscope with integrated OMEGA energy filter, operated at 80 kV in the zero-loss mode (Zeiss, https://www.zeiss.com). Images were acquired with a 2k × 2k slow-scan CCD camera (TRS Tröndle Restlichtverstärkersysteme, Moorenweis, Germany).

## Ribosome co-sedimentation assay

Plants were densely pot-grown either 4 weeks at normal conditions or transferred to 4 °C after 3 weeks for 21 days prior to harvesting. The plant material was cryogenically ground, and ribosomes were enriched from 250 mg material. First, the material was resuspended in lysis buffer (50 mM HEPES pH 8, 25 mM KCl, 10 mM $MgCl_2$, 100 µg/mL chloramphenicol, 100 µg/mL cycloheximide, 1 mM DTT, 0.2 mg/mL heparin, 1× EDTA-free cOmplete protease inhibitor (Roche), 1% dodecylmaltoside, 10 mM ribonucleoside vanadyl complex (NEB)) by a stainless-steel potter (Carl Roth) in 2 mL tubes and solubilized by a 10 min incubation at 4 °C with end-over-end shaking. Cell debris were cleared by centrifugation at $16,000 \times g$ for 15 min at 4 °C and 400 µL were layered on a 750 µL 25% sucrose cushion (prepared in lysis buffer without DDM and ribonucleoside vanadyl complex). After centrifugation for 20 min at $173,292 \times g$ in a MLA-130 rotor, the pellet was resuspended in 80 µL lysis buffer. For mass spectrometry, 10 µL of the resuspended pellet were separated 1 cm with a 10% SDS-PAGE. Peptides generated by in-gel digest from three gel slices[28] were measured as described below.

## Proteomic analyses

For the whole-cell proteomics, plants (Col-0, *tig2*) were densely grown in pots. For cold treatment, plants were transferred to 4 °C, two weeks after germination and kept for 3 weeks. Control plants remained for additional two weeks under same light and temperature conditions. 150 mg of plant material was used per experiment and the $n = 4$ independently grown biological replicates, technical replicates were not performed. After lysis by cryo-genic grinding, samples were thawed by the addition of 250 µL lysis buffer (100 mM $NaCO_3$, 1× Roche Complete EDTA-free Protease inhibitor) and the solution was mixed in a Retsch MM400 at 27 hz for 1 min in ice-cold Teflon tube-adaptors using 2 4 mm stainless steel balls. Samples were denatured by addition of 4% SDS (w/v) with 20% sucrose (w/v) buffer and boiled for 1 min at 96 °C. The concentration was adjusted via BCA (Pierce) and 50 µg of total protein was loaded on a 10% SDS-PAGE. Samples were run for approximately 1.5 cm into the separating gel, gels were stained with colloidal Coomassie and the lanes were split into three equal slices for in-gel digestion with trypsin as described before[28]. Peptides were separated on an EASY-nLC™ 1200 chromatography system (Thermo Scientific) coupled to a Q Exactive HF Orbitrap LC-MS/MS System (Thermo Fisher Scientific) via a Nanoflex ion source. Analytical columns (50 cm, 75 µm inner diameter packed in-house with C18 resin ReproSilPur 120, 1.9 µm diameter Dr. Maisch) were run at a flow rate of 250 nL/min using buffer A (aqueous 0.1% formic acid) and buffer B (80 % acetonitrile, 0.1% formic acid). MS data was acquired in data dependent fashion using a Top15 method. Full scans (300–1650 m/z, $R = 60,000$ at 200 m/z) were acquired at a target of 3e6 ions and a maximum injection time of 20 ms. The fifteen most intense ions were isolated and fragmented with higher-energy collisional dissociation (HCD) (target 1e5 ions, maximum injection time 80 ms, isolation window 1.4 m/z, NCE 28%) and detected in the Orbitrap ($R = 15,000$ at 200 m/z). Additional parameters for chromatography and MS instrument settings can be retrieved from the raw files available at ProteomeXchange.

The raw data were processed with MaxQuant (2.0.1.0) using the 2022 Uniprot Arabidopsis proteome fasta file. A false-discovery rate (FDR) of 0.01 for peptides and for proteins and a minimum peptide length of 7 amino acid residues were required. For Andromeda search, trypsin allowing for cleavage N-terminal of proline was chosen as enzyme specificity. Cysteine carbamidomethylation was selected as a

fixed modification, and protein N-terminal acetylation and methionine oxidation were selected as variable modifications. A maximum of two missed cleavages was permitted. Initial mass deviation of precursor ion was limited to 7 ppm and mass deviation for fragment ions to 0.5 Da. Protein identification required at least one unique in each protein group. "Match between run" was used to transfer identities within all replicate samples. Data analysis was performed with Perseus[52]. In brief, proteins were removed that were identified in less than three out of four replicates in one of the four categories (Col-0 at 21 °C or 4 °C; *tig2 at* 21 °C or 4 °C), the LFQ-values were $Log_2$-transformed and enrichment between the lines and conditions was determined by two-sided *t*-test and permutation-based FDR of 0.05 and $S_0 = 1$ (when both compared sample sets contained at least 3 valid values). Identifiers of proteins with significantly altered abundance were used for GO enrichment analysis using the Panther website[32]. For enrichment analyses, missing values were not imputed. Instead, proteins that were absent in all replicates of one line (mutant or Col-0, respectively) were marked, all other proteins with missing values were removed from further analysis (Supplementary Data 1). Significantly enriched proteins and the marked proteins were all considered in further data evaluation.

### AlphaFold prediction of ribosome interaction

Structural predictions of Tig2's interaction with chloroplast ribosomes were modelled using the AlphaFold 3 algorithm via the online server (www.alphafoldserver.com; October 2024). Briefly, amino acid sequences of mature Tig1 (At5g55220), Tig2 (At2g30695), uL23c (AtCg01300), uL29c (At5g65220), uL24c (At5g54600), uL32c (AtCg01020), uL22c (AtCg00810) were modelled together with the RNA sequence of 23s-rRNA. For nuclear encoded sequences, transit peptides were removed beforehand. Predictions were repeated twice on different seeds (12345678, 87654321), with Tig2 being consistently observed at the exit tunnel in all obtained models.

### Immunolocalization

Protoplasts were prepared from Col-0 grown for 17 days under long-day conditions with a light intensity of 120 μM (16 h light/8 h dark). Protoplasts were isolated according to the protocol by Yoo et al.[53]. Protoplast pre-treatment and immunofluorescence analysis (IFA) were conducted as described[54]. For labeling, 1 μg of the Tig2 antibody and IgG Rabbit antibody (Thermo Fisher, Germany), respectively, were labeled using the Flexible Coralite 488 Antibody Labeling Kit according to the manufacturer's instructions (Proteintech, Germany) and used at the 1:250 dilution in the hybridization buffer[54]. Images were captured using the Abberior Confocal Microscope (Abberior, Germany) using a 488 nm excitation laser. Additionally, the same laser was used to capture images of the background autofluorescence within the detection range of 580–650 nm. A permanent liquid mountant with DAPI (Vectashield, USA) was used to stain the nuclei and nucleoids of chloroplasts. DAPI images were obtained by excitation with a 405 nm laser. Imaging of DAPI, 488 nm, and autofluorescence channels was conducted simultaneously.

### Miscellaneous

Photosynthetic measurements were done as published previously[12]. In the Northern blot assay, a total of 1 μg of cellular RNA was separated on a 1.2% agarose gel. A membrane stain with Midori Green was performed as a loading control. The membrane was hybridized with strand-specific RNA probes generated against 23S and 16S RNA fragments covering both mature and unprocessed 23S RNAs. Additionally, a specific fragment covering a 1.1 kb segment of the 23S rRNA was generated. Probe synthesis was based on PCR fragments carrying the T7 promoter sequences. The probes were labelled with Cy7.5 fluorophore for the 23S rRNA and Cy5.5 for the 16S RNA fragment. RNA probes were subsequently hybridized overnight at 65 °C and washed four times for 15 min each at the hybridization temperature

with 0.1× SSC; 0.1% SDS and 0.05× SSC; 0.1% SDS, respectively. The signal was detected by scanning the membrane on a Licor Dlx scanner with automatic setup. Polysome analyses were done as published[28,55] with slight modifications. In brief, 200 mg of 3–4 weeks old leaf material was ground and extracted in 1 mL buffer (50 mM HEPES-KOH ph 8.0, 200 mM KOAc, 35 mM MgOAc, 25 mM EGTA, 1% Triton X-100, 2% polyoxyethylene (10) tridecyl ether, 200 mM sucrose, 100 mM β-mercaptoethanol, 100 μg/mL Chloramphenicol, 25 μg/mL Cycloheximid, 0.5 mg/ml Heparin, 100U SUPERase·In™ RNase Inhibitor (Invitrogen, Thermo Fisher Scientific) and protease inhibitor cOmplete™ EDTA-free Protease Inhibitor Cocktail, Roche). Debris was removed by centrifugation and microsomal membranes were solubilized by addition of 0.5% (v/v) Sodium deoxycholate. Fractions were collected from a 15 to 55% sucrose gradient after centrifugation at 275.500 × *g* for 110 min in a Beckman SW 41 rotor at 275.500 × *g* for 110 min. Fractions were precipitated by Methanol/Chloroform, prior to loading on an SDS-PAGE. Protein extraction and immunoblotting was carried out as described before[56]. Immunoblots were captured with the Intas ChemoStar. All used antibodies are listed in Table S3.

### Statistics & reproducibility

All experiments were conducted with at least three independent biological replicates, proteomic experiments were conducted with four biological replicates. Plants position under the light filed was randomized, however, the investigators were not blinded to allocation during experiments and outcome assessment. Statistical analyses of the individual data are given in the Figure legends.

### Reporting summary

Further information on research design is available in the Nature Portfolio Reporting Summary linked to this article.

## Data availability

Proteomics data are available via ProteomeExchange consortium (http://www.ebi.ac.uk/pride) with the following identifier: PXD057252 Data Set 1 - Global proteome of Col-0 and *tig2* mutant [https://www.ebi.ac.uk/pride/archive/projects/PXD057252] PXD057264 Data Set 2– Analysis of sedimented ribosomes [https://www.ebi.ac.uk/pride/archive/projects/PXD057264] All gene accession numbers are given in Supplementary Data 1 and are according to the Arabidopsis standard https://www.arabidopsis.org. Source data are provided with this paper.

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

## Acknowledgements

We thank Karin Gries for help with protein purification, Michael Schroda, Vincent Gotsmann, Claudia Herkt, Peter Emelin and Günter Kramer for critical discussion of the data. Thanks to Sandro Keller for providing devices for CD and DLS measurements. This work was supported by the Carl-Zeiss fellowship to F.R., the Deutsche Forschungsgemeinschaft grant TRR175 A01 (C.S.L), A05 (F.W.) and B08 (T.M.) and WI 3477/3-1 to F.W. and the Forschungsschwerpunkt BioComp to F.W., D.S., T.M., and M.R.

## Author contributions

F.R. performed and/or designed most of the experiments and wrote parts of the manuscript, J.G., S.N., S.K., V.S., H.P., and T.M. performed phenotypic analyses and created Arabidopsis lines. C.S. and A.K. conducted electron microscopy analyses and analyzed data. J.L. and C.S.L. performed northern blot analyses and immunofluorescence. J.Z.Y.N., D.W., and G.K.A.H performed phylogeny and part of the AlphaFold prediction. M.R. did proteomic measurements. D.S. helped with mutant analyses, line generation and performed microscopy. F.W. designed experiments, analyzed data, and wrote the manuscript.

## Funding

## Competing interests

The authors declare no competing interests.
