## [Peer Review File · Nature Communications]

REVIEWER COMMENTS

Reviewer #1 (Remarks to the Author):

This study presents a truncated variant of the classical ribosome-associated trigger factor, named Tig2, found in plastids of *Arabidopsis thaliana* that associates with chlororibosomes. Phylogenetic analysis shows that Tig2 is a result of gene duplication of an ancestral gene before the divergence of chlorophytes and streptophytes, and was subsequently lost in most chlorophytes. Unlike typical trigger factors, Tig2 lacks chaperone activity, which the authors demonstrated with a DLS assay. Under cold conditions, its deletion results in developmental defects, impaired photosynthesis, and impaired chlororibosome biogenesis, manifested by rRNA cleavage. The TEM analysis of leaf cross-sections is very nice, and the proteome analysis convincingly shows Tig2's role in chlororibosome association. Mass spectrometric quantification of ribosomal pellets in *tig2* mutants showed enrichment of RH39 helicase that facilitates breaks in the rRNA. The authors conclude that Tig2 is essential for biogenesis and has specifically evolved to protect the exit tunnel during chlororibosomal maturation from rRNA breaks. Overall, the authors did an excellent job in identifying and characterizing Tig2, and the paper is clear.

I have no comments on the technical aspects of the performed studies but only some suggestions on how to improve the figures and analysis. In my opinion, what is really missing in the paper is the structural component, and at the moment it reads a bit unfinished. The authors should at least try some of the available *in silico* tools combined with previous studies to complement their work and provide more molecular details about Tig2 structure and function, including its binding to the ribosome.

1. Figure 1d: If you swap the two models, then the N-domains will be next to each other for easier comparison. Labeling the domains on the models would also help. Additionally, the structure of Chlamy Tig1 reported by the authors (PMID: 36189745) and cited in this paper looks different from the shown model.
2. Figure 7d: On the overview, add labels for the defining features of the ribosome so that the reader can easily identify the orientation. The zoomed-in panel can also illustrate the tunnel, which is hardly visible at the moment. For the structural analysis of the chlororibosome, please use the most accurate and updated coordinates, PDB ID 6ERI.
3. Abstract: In the last sentence of the abstract, the authors state, "Tig2 illustrates a fascinating concept of how a chaperone's domain evolved as an individual protein, serving a completely different task." I share the excitement; however, the discovery should also be put in the context of earlier findings of similar phenomena related to ribosomal tunnel exit-binding proteins. For example, a protein homologous to the M-domain of the bacterial SRP binding protein Ffh is found in its ribosome-bound form in proximity to bL23m (PMID: 32553108, PMID: 36253367). These evolutionary intermediates in the exit tunnel are analogous to the presented story and should be discussed somewhere.

4. Page 19, Line 414: The authors suggest that Tig2 might interact with fully assembled ribosomal complexes. Therefore, an effort should be made to model Tig2 on the ribosome using conventional in silico tools. For example, the AlphaPulldown program offers a high-throughput in silico protein-protein interaction screen, which can be combined with a database of potential candidates. And since the ribosome-interacting interface of Tig2 can be easily derived from the known structures, the model with the chlororibosome can be predicted and presented with a high level of confidence and discussed in the functional context. One idea is to compare it with the mitoribosome-insertase complex (6ZM5) and discuss in the context of membrane-mediation.

Reviewer #2 (Remarks to the Author):

The authors of the manuscript present the first characterization of a second putative trigger factor-like protein, Tig2, that is widely present in green plants, specifically in streptophytes. Tig2 is a chloroplast localized protein, that resembles the ribosome binding domain of Tig1, the canonical ribosome-binding chaperone trigger factor.

Using a variety of molecular biology, biophysical, and biochemical techniques, the authors describe that under cold conditions, *Arabidopsis thaliana* Tig2 loss of function mutants have reduced photosynthesis and mild morphological phenotypes at the ultrastructural level. Furthermore, the authors provide compelling observations about the possible implication of Tig2 in ribosome biogenesis, specifically affecting ribosomal proteins abundances and rRNA processing. These findings are unexpected and relevant to the field, and provide novel insights into the potential involvement of molecular chaperones in ribosome assembly. Considering that the study does not provide any mechanistic insights and a potential role of Tig2 as a nascent chain chaperone is not fully explored, in its current state we would support publication only if the following concerns are addressed:

1. The title states “A truncated variant of the ribosome-associated trigger factor specifically assists chloroplast ribosome biogenesis in plants”, the word “assists” should be replaced, as the investigation does not provide enough evidence to claim a direct role for Tig2 in ribosome biogenesis. The word “contributes” may be more suitable.

2. For figure S2, please add the polar and non-polar residue distribution comparison of Tig2 and Tig1 of *Arabidopsis thaliana*. Such differences can aid the reader picture its novel functional role.

3. While we agree that Tig2 fails to prevent aggregation of RbcL and GAPDH, it is not correct to assume that a fully synthesized RbcL or GAPDH are suitable substrates. The correct substrates of Tig2 may be nascent chains or a completely different pool of proteins. The phrasing of the text should therefore be more moderate and state that using this substrate/assay, a chaperone function could not be detected. Accordingly, the text in the discussion section should be modified.

4. Further, we believe the potential of Tig2 to act as a chaperone of nascent chains has not been fully explored and it might aid explaining some of the mass spectrometry based proteomic analysis found later on the manuscript. To this end, we suggest analyzing Tig2 co-migration with polysomes derived from chloroplasts by means of sucrose gradients. Such assay could help pinpoint a potential role during translation and nascent chain maturation.

5. Along the same line of concern #4, why would Tig2 co-sediment more strongly with ribosomes than Tig1?

6. Considering that nucleoids have been suggested to be at least partly involved in ribosome biogenesis, the possible nucleoid localization of overexpressed Tig2 is a very intriguing observation. It however is puzzling that the phenomenon is visible only in some cases. The authors need to demonstrate that this phenomenon is not caused by overexpression. Ideally, nucleoid localization should be demonstrated by native immunohistochemistry of Tig1 and Tig2 and not heterologous expression. Concomitantly, this experiment should be performed with proteins known to localize in the nucleoid.

7. Figure 3E and G show remarkable phenotypes as shown in the data in bar plots (bottom figure 3E), but the top panels should show Col-0 and *tig2* deletion for evaluating the maximum quantum yield of PSII and the images of young leaves, not the *tig1/2* double mutant. *Tig1/2* double mutant accordingly, should then be moved to the supplement.

8. The authors state that 13 ribosomal proteins were downregulated and 9 of those are part of the large ribosomal subunit in *tig2* deletion. In order to come up with some hints towards the molecular mechanism involved, the authors should perform additional analyses of these affected proteins, by exploring what unifying features these may have, e.g. position on the ribosome, fold, size, plastid-encoded or nuclear-encoded, etc.

9. The sentence starting in line 311 “Indeed, our sucrose cushion assays of Col-0 samples from seedlings at 21°C showed that Tig2 co-sediments with ribosomes in a high-salt-sensitive and puromycin-sensitive manner” is wrongly stated and very confusing. Co-sedimentation is in fact high-salt-sensitive; however, it is puromycin-insensitive.

10. Eleven ribosomal proteins of the small subunit of the plastid ribosome were significantly affected when analyzed from mass spectrometry of ribosomal particles. How many of those were downregulated? Can the authors provide a model explaining how small ribosomal subunit proteins are affected, if the binding site of Tig2 is in the large ribosomal subunit? This should be included into the discussion.

11. The sentence in line 93 should be: “Hidden breaks are introduced by rRNA cleavage in a post-maturation step of ribosome biogenesis.”

Reviewer #3 (Remarks to the Author):

Reviewer #4 (Remarks to the Author):

The manuscript “A truncated variant of the ribosome-associated trigger factor specifically assists chloroplast ribosome biogenesis in plants” by Ries et al. assigns a novel functionality to Tig2, a truncated ortholog of the ribosome-associated chaperone trigger factor Tig1.

The authors present a very well structured and systematic study to show that Tig2 most likely evolved an independent function from Tig1 in ribosome biogenesis. This novel functionality of Tig2 has not been described before and is a valuable addition to the knowledge about chloroplast ribosome biology. This is solid piece of work with well-performed experiments and a very clear presentation of the data. I would recommend publication with major revision of the proteomic data presentation.

Major comments:

1) Proteomics dataset1 (whole cell). You performed the statistical comparison between groups without data imputation (Supplemental Data, Tab-whole cell). Proteins that are identified in only

one group therefore do not appear in the Volcano plot depiction. This also includes Tig2 itself, which as expected is only detected in the WT but not in the tig2 mutant. Could you explain why you chose not to use imputation at all? Even though the information is available in the Supplemental table, it would be useful for a broader audience to include the information about proteins specific for a group or genotype more clearly in the manuscript. Did you consider these group-specific proteins in your interpretation of the proteomic results?

2) The Tig1 and Tig2 measurements for Col-4° and Col-21°C from Proteomics dataset1 (whole cell) should be linked to your Transcriptome and Western Blot measurement results from Figure 2c,d. For Tig2 you detect a significantly higher protein amount at 4°C vs 22°C in the proteomics measurements.

3) Proteomics Data upload. Please correct the proteomics data upload for Proteomics Dataset 1: PXD052583. In the currently uploaded txt SEARCH file the experiment annotation for the raw files is incorrect. The groups tig2_RT and Col_4C have been swapped. According to your Supplemental data table raw file IDs P0142_13-24 should refer to Col_4C and IDs P0142_25-36 to tig2_RT. The MaxQuant version for the search was 2.0.1.0 and not 1.6.3.3 as referenced in the manuscript method section and the PRIDE method description. Please correct.

4) Proteomics Data upload. Please correct the proteomics data upload for Proteomics Dataset 2: PXD052586. The currently uploaded txt SEARCH file was searched against the fasta file for Chlamydomonas (CreinhardtiiCC_4532_707_v6.1.protein_cleaned.fasta). The MaxQuant version for the search was 2.0.1.0 and not 1.6.3.3 as referenced in the manuscript method section and the PRIDE method description. Please correct.

5) Since the PRIDE upload contains the search against Chlamydomonas and not Arabidopsis this could not be confirmed but from the Supplemental table (Dataset 2, tab-ribo pellet 4C, ribo pellet 21C) it appears you filtered the ribosome dataset for proteins with 3 quantitative values in ALL groups. Do you have a rationale for this very strict filtering? With this you loose also all information about proteins specific for one of the conditions and it might explain, why TIG2 is not present in the proteomics data table for dataset 2 even though it was identified as being present in the ribosomal pellet in the Western blot analysis (Fig 7a). As for dataset 1, please comment if the inclusion of group-specific proteins alters your interpretation of the proteomic results.

6) Figure S10 a. You show that tig1 mutants have more starch granules at 4°C than the Wild type. Can you explain the phenotype for the double mutant tig1/2 which shows a clear reduction in starch granule number even though this cannot be observed for the single mutants?

Minor comments

LINE 187, Figure 3c,d: Could you add the datapoints to the bar plot depiction for Figure 3c and d? The differences you mention in the text look very marginal in the bar plot.

LINE 828: Figure 3 c text. Scale war -> should be "bar"

LINE 829: Figure 3 d text: Roth length -> should be "Root"

Supplemental dataset. Tab-whole cell. The sample names contain for room temperature the value 21°C. In the statistical comparison columns the naming contains 22°C. e.g. p-value(-log10)_LFQ_tig2_22°C_vs_LFQ_CoI-0_22°C_S_01_FDR_0.05

We thank all four reviewers for their time and the highly valued suggestions to improve this manuscript. Below, we explain how we added the data and how we addressed the specific concerns. All changes are marked in red within the revised manuscript.

Response to reviewers

This study presents a truncated variant of the classical ribosome-associated trigger factor, named Tig2, found in plastids of *Arabidopsis thaliana* that associates with chlororibosomes. Phylogenetic analysis shows that Tig2 is a result of gene duplication of an ancestral gene before the divergence of chlorophytes and streptophytes, and was subsequently lost in most chlorophytes. Unlike typical trigger factors, Tig2 lacks chaperone activity, which the authors demonstrated with a DLS assay. Under cold conditions, its deletion results in developmental defects, impaired photosynthesis, and impaired chlororibosome biogenesis, manifested by rRNA cleavage. The TEM analysis of leaf cross-sections is very nice, and the proteome analysis convincingly shows Tig2's role in chlororibosome association. Mass spectrometric quantification of ribosomal pellets in *tig2* mutants showed enrichment of RH39 helicase that facilitates breaks in the rRNA. The authors conclude that Tig2 is essential for biogenesis and has specifically evolved to protect the exit tunnel during chlororibosomal maturation from rRNA breaks. Overall, the authors did an excellent job in identifying and characterizing Tig2, and the paper is clear.

I have no comments on the technical aspects of the performed studies but only some suggestions on how to improve the figures and analysis. In my opinion, what is really missing in the paper is the structural component, and at the moment it reads a bit unfinished. The authors should at least try some of the available *in silico* tools combined with previous studies to complement their work and provide more molecular details about Tig2 structure and function, including its binding to the ribosome.

1. Figure 1d: If you swap the two models, then the N-domains will be next to each other for easier comparison. Labeling the domains on the models would also help. *The two models are now swapped showing Tig2 first. Domains are also labelled now.* Additionally, the structure of Chlamy Tig1 reported by the authors (PMID: 36189745) and cited in this paper looks different from the shown model. *In fact, we have compared the alpha fold model of Tig1 with our previously published SAXA model and the Chlamydomonas crystal structure of the PPlase and the chaperone domain. Based on this, the domain architecture of the predictions appears realistically, although the domain orientation might vary (this may be demonstrating the flexible nature of the Tig1 molecule). We added a short sentence to the text (line 128-132) and Tig1 AlphaFold prediction and the SAXS-based model are compared in Fig. S2c.*

2. Figure 7d: On the overview, add labels for the defining features of the ribosome so that the reader can easily identify the orientation. The zoomed-in panel can also illustrate the tunnel, which is hardly visible at the moment. For the structural analysis of the chlororibosome, please use the most accurate and updated coordinates, PDB ID 6ERI. *We have replaced the panel (now Figure 7e) with the ribosome features, based on the coordinates of PDB ID 6ERI. We also included additional labels to better highlight the important features of the ribosome exit site.*

3. Abstract: In the last sentence of the abstract, the authors state, "Tig2 illustrates a fascinating concept of how a chaperone's domain evolved as an individual protein, serving a completely different task." I share the excitement; however, the discovery should also be put in the context of earlier findings of similar phenomena related to ribosomal tunnel exit-binding proteins. For example, a protein homologous to the M-domain of the bacterial SRP binding protein Ffh is found in its ribosome-bound form in proximity to bL23m (PMID: 32553108, PMID: 36253367). These evolutionary intermediates in the exit tunnel are analogous to the presented story and should be discussed somewhere. *Thank you, for the*

fantastic thought. We added the information and the idea at the end of the discussion (lines 464-470).

4. Page 19, Line 414: The authors suggest that Tig2 might interact with fully assembled ribosomal complexes. Therefore, an effort should be made to model Tig2 on the ribosome using conventional in silico tools. For example, the AlphaPulldown program offers a high-throughput in silico protein-protein interaction screen, which can be combined with a database of potential candidates. And since the ribosome-interacting interface of Tig2 can be easily derived from the known structures, the model with the chlororibosome can be predicted and presented with a high level of confidence and discussed in the functional context. One idea is to compare it with the mitoribosome-insertase complex (6ZM5) and discuss in the context of membrane-mediation. In order to address this concern, we used the AlphaFold 3 server to model Tig2 binding to the 50S section surrounding the tunnel. To this end, we tested different combinations including the ribosomal 23S rRNA, the ribosomal proteins uL22c, uL23c, uL24c, uL29c, uL32C with both Tig1 and Tig2 or just with Tig2. For each combination, 5 predictions were performed. Importantly, all predictions that included Tig2 modeled this protein adjacent to the tunnel exit site and it seems most likely that Tig2 interacts with uL24C. The data are now added as new panel “b” to Figure 7. The results section was complemented accordingly (lines 331 to 344).

Reviewer #2 (Remarks to the Author):

The authors of the manuscript present the first characterization of a second putative trigger factor-like protein, Tig2, that is widely present in green plants, specifically in streptophytes. Tig2 is a chloroplast localized protein, that resembles the ribosome binding domain of Tig1, the canonical ribosome-binding chaperone trigger factor.

Using a variety of molecular biology, biophysical, and biochemical techniques, the authors describe that under cold conditions, *Arabidopsis thaliana* Tig2 loss of function mutants have reduced photosynthesis and mild morphological phenotypes at the ultrastructural level. Furthermore, the authors provide compelling observations about the possible implication of Tig2 in ribosome biogenesis, specifically affecting ribosomal proteins abundances and rRNA processing. These findings are unexpected and relevant to the field, and provide novel insights into the potential involvement of molecular chaperones in ribosome assembly. Considering that the study does not provide any mechanistic insights and a potential role of Tig2 as a nascent chain chaperone is not fully explored, in its current state we would support publication only if the following concerns are addressed:

1. The title states “A truncated variant of the ribosome-associated trigger factor specifically assists chloroplast ribosome biogenesis in plants”, the word “assists” should be replaced, as the investigation does not provide enough evidence to claim a direct role for Tig2 in ribosome biogenesis. The word “contributes” may be more suitable. **Changed**

2. For figure S2, please add the polar and non-polar residue distribution comparison of Tig2 and Tig1 of *Arabidopsis thaliana*. Such differences can aid the reader picture its novel functional role. **Data are added.**

3. While we agree that Tig2 fails to prevent aggregation of RbcL and GAPDH, it is not correct to assume that a fully synthesized RbcL or GAPDH are suitable substrates. The correct substrates of Tig2 may be nascent chains or a completely different pool of proteins. The phrasing of the text should therefore be more moderate and state that using this substrate/assay, a chaperone function could not be detected. Accordingly, the text in the discussion section should be modified. **We adjusted the text in the Results (lines 151-153) and the Discussion (lines 391-393).**

4. Further, we believe the potential of Tig2 to act as a chaperone of nascent chains has not been fully explored and it might aid explaining some of the mass spectrometry based proteomic analysis found later on the manuscript. To this end, we suggest analyzing Tig2 co-migration with polysomes derived from chloroplasts by means of sucrose gradients. Such assay could help pinpoint a potential role during translation and nascent chain maturation. In fact, we do not believe that Tig2 act as chaperone of nascent chains in chloroplasts but rather promotes ribosome biogenesis. But we agree that polysome analyses is an important addition that will help to interpret the data. We performed polysome analyses, as requested by the reviewer. The data are now included in Supplemental Figure S14.

5. Along the same line of concern #4, why would Tig2 co-sediment more strongly with ribosomes than Tig1? This finding is actually not contradictory, assuming that Tig2 has a comparable function as the cytosolic, eukaryotic ribosome placeholder EBP1/Arx1. When cytosolic ribosomes were isolated for Cryo-EM or protomic analyses, EBP1 is abundantly associated with ribosomes (almost at equimolar levels as ribosomal proteins), which is far higher than other co-translationally associated proteins (e.g. Kraushar et al., Molecular Cell, 2021; Wells et al., PLOS Biology, 2020). Thus, higher amounts of co-sedimenting Tig2 would be in agreement with the observations on EBP1. We added a short sentence to the discussion, to clarify this (lines 445-448).

6. Considering that nucleoids have been suggested to be at least partly involved in ribosome biogenesis, the possible nucleoid localization of overexpressed Tig2 is a very intriguing observation. It however is puzzling that the phenomenon is visible only in some cases. The authors need to demonstrate that this phenomenon is not caused by overexpression. Ideally, nucleoid localization should be demonstrated by native immunohistochemistry of Tig1 and Tig2 and not heterologous expression. Concomitantly, this experiment should be performed with proteins known to localize in the nucleoid. Thank you for the suggestion. We performed additional immunofluorescence microscopy with Arabidopsis protoplasts and could not detect comparable foci as seen in the Tig2-GFP line. We thus think that the foci might result from overaccumulation of Tig2-GFP. We modified the text accordingly (lines 165-168).

7. Figure 3E and G show remarkable phenotypes as shown in the data in bar plots (bottom figure 3E), but the top panels should show Col-0 and *tig2* deletion for evaluating the maximum quantum yield of PSII and the images of young leaves, not the *tig1/2* double mutant. *tig1/2* double mutant accordingly, should then be moved to the supplement. We now replaced the pannels with images comparing Col-0 and *tig2* during cold acclimation. The previous images of Col-0 versus *tig1/2* were moved to Supplemental Figure 8a.

8. The authors state that 13 ribosomal proteins were downregulated and 9 of those are part of the large ribosomal subunit in *tig2* deletion. In order to come up with some hints towards the molecular mechanism involved, the authors should perform additional analyses of these affected proteins, by exploring what unifying features these may have, e.g. position on the ribosome, fold, size, plastid-encoded or nuclear-encoded, etc. We revisited these data and realized that the phrasing in the Results might have been a bit misleading. In fact, most ribosomal proteins of both subunits were reduced in the proteomics data of cold-treated *tig2*. Compared to proteins of the small subunit, proteins of the large subunit were slightly more affected and more proteins were significantly reduced. In order to address this comment, we assayed if the significantly reduced proteins have particular features, however, we could not observe features that stood out for these proteins. Thus, we decided not to mention this in the text. We rather intended to explain that ribosome accumulation is generally affected by the absence of Tig2 and did not want to draw the focus on individual proteins. We rephrased the text in lines 279-283 and hope that it is less distracting now.

9. The sentence starting in line 311 "Indeed, our sucrose cushion assays of Col-0 samples from seedlings at 21°C showed that Tig2 co-sediments with ribosomes in a high-salt-

sensitive and puromycin-sensitive manner” is wrongly stated and very confusing. Co-sedimentation is in fact high-salt-sensitive; however, it is puromycin-insensitive. We now added the p-values for the quantification of the Puromycin treatment (Tig1 and Tig2). Since the puromycin treatment did not significantly change the ribosome association, we changed the sentence as requested by the reviewer (lines 327-329).

10. Eleven ribosomal proteins of the small subunit of the plastid ribosome were significantly affected when analyzed from mass spectrometry of ribosomal particles. How many of those were downregulated? Can the authors provide a model explaining how small ribosomal subunit proteins are affected, if the binding site of Tig2 is in the large ribosomal subunit? This should be included into the discussion. We added a respective section to the Discussion (lines 471-478).

11. The sentence in line 93 should be: “Hidden breaks are introduced by rRNA cleavage in a post-maturation step of ribosome biogenesis.” Changed (now line 95-96).

Reviewer #3 (Remarks to the Author):

Reviewer #4 (Remarks to the Author):

The manuscript “A truncated variant of the ribosome-associated trigger factor specifically assists chloroplast ribosome biogenesis in plants” by Ries et al. assigns a novel functionality to Tig2, a truncated ortholog of the ribosome-associated chaperone trigger factor Tig1. The authors present a very well structured and systematic study to show that Tig2 most likely evolved an independent function from Tig1 in ribosome biogenesis. This novel functionality of Tig2 has not been described before and is a valuable addition to the knowledge about chloroplast ribosome biology. This is solid piece of work with well-performed experiments and a very clear presentation of the data. I would recommend publication with major revision of the proteomic data presentation.

Major comments:

1) Proteomics dataset1 (whole cell). You performed the statistical comparison between groups without data imputation (Supplemental Data, Tab-whole cell). Proteins that are identified in only one group therefore do not appear in the Volcano plot depiction. This also includes Tig2 itself, which as expected is only detected in the WT but not in the *tig2* mutant. Could you explain why you chose not to use imputation at all? Even though the information is available in the Supplemental table, it would be useful for a broader audience to include the information about proteins specific for a group or genotype more clearly in the manuscript. Did you consider these group-specific proteins in your interpretation of the proteomic results? Initially, we performed imputation of missing values, consistent with our previous proteomics studies (Westrich et al., NAR, 2021 and Trösch et al., Plant Cell, 2022). Data imputation adds low abundant and random values for the missing data. However, we realized that imputation generated the impression that several low abundant proteins, including Tig2 and ribosome biogenesis factors, were statistically indifferent and accumulated to equal amounts in Col-0 and the mutant. The phenomenon was observed for different imputation parameters. We thus decided to avoid imputation in that specific case and rather categorized these proteins in the Supplemental Dataset (“Col-0 only” and “*tig2* only”, respectively). We only considered proteins for these categories, if all four replicates had missing values in Col-0 or *tig2*. Although we could not give enrichment values for these

proteins and had to exclude them from the volcano plot, we considered these proteins for subsequent analyses, such as the heat map and the GO-term analysis. We now explained this procedure better in the revised Methods (lines 612-617).

2) The Tig1 and Tig2 measurements for Col-4° and Col-21°C from Proteomics dataset1 (whole cell) should be linked to your Transcriptome and Western Blot measurement results from Figure 2c,d. For Tig2 you detect a significantly higher protein amount at 4°C vs 22°C in the proteomics measurements. We now linked these findings in the Results section (lines 259-261).

3) Proteomics Data upload. Please correct the proteomics data upload for Proteomics Dataset 1: PXD052583. In the currently uploaded txt SEARCH file the experiment annotation for the raw files is incorrect. The groups tig2_RT and Col_4C have been swapped. According to your Supplemental data table raw file IDs P0142_13-24 should refer to Col_4C and IDs P0142_25-36 to tig2_RT. The MaxQuant version for the search was 2.0.1.0 and not 1.6.3.3 as referenced in the manuscript method section and the PRIDE method description. Please correct. Indeed, thank you for bringing this to our attention. Uploads and MaxQuant version were corrected.

4) Proteomics Data upload. Please correct the proteomics data upload for Proteomics Dataset 2: PXD052586. The currently uploaded txt SEARCH file was searched against the fasta file for Chlamydomonas (CreinhardtiiCC_4532_707_v6.1.protein_cleaned.fasta). The MaxQuant version for the search was 2.0.1.0 and not 1.6.3.3 as referenced in the manuscript method section and the PRIDE method description. Please correct. The uploads and MaxQuant version description were corrected.

5) Since the PRIDE upload contains the search against Chlamydomonas and not Arabidopsis this could not be confirmed but from the Supplemental table (Dataset 2, tab-ribo pellet 4C, ribo pellet 21C) it appears you filtered the ribosome dataset for proteins with 3 quantitative values in ALL groups. Do you have a rationale for this very strict filtering? With this you lose also all information about proteins specific for one of the conditions and it might explain, why TIG2 is not present in the proteomics data table for dataset 2 even though it was identified as being present in the ribosomal pellet in the Western blot analysis (Fig 7a). As for dataset 1, please comment if the inclusion of group-specific proteins alters your interpretation of the proteomic results. We apologize for the confusion with the upload of a wrong dataset. To clarify how we filtered: we filtered for three values in the lines of a respective temperature condition and not for all groups, this would have been certainly too strict. We were actually surprised not to find Tig2 in the ribosomal pellet of the Col-0 samples. In these samples, maybe Tig2 was covered by highly abundant ribosomal proteins. Concerning the last point, we wanted to be consistent with our previous analysis and considered only proteins for statistical analysis that were present in at least three out of four replicates. Nevertheless, we checked the data for this concern again and found only a very minor fraction of proteins specifically in one group, of which most had no predicted chloroplast localization. Thus, we did not further mention this particular group in the text.

6) Figure S10 a. You show that tig1 mutants have more starch granules at 4°C than the Wild type. Can you explain the phenotype for the double mutant tig1/2 which shows a clear reduction in starch granule number even though this cannot be observed for the single mutants? Our immunoblots in Figure 6 revealed that RbcL accumulates at lowest levels when both trigger factor variants are absent. Thus, the double mutant might have an impaired Calvin Benson Cycle in the cold, causing the strong reduction of starch granules in chloroplasts of the double mutant. We now linked these two findings in the result section (lines 304-306).

Minor comments:

LINE 187, Figure 3c,d: Could you add the datapoints to the bar plot depiction for Figure 3c and d? The differences you mention in the text look very marginal in the bar plot. Data points are now added to Fig. 3c,d and Fig S7.

LINE 828: Figure 3 c text. Scale war -> should be "bar" 😊 Changed

LINE 829: Figure 3 d text: Roth length -> should be "Root". Changed

Supplemental dataset. Tab-whole cell. The sample names contain for room temperature the value 21°C. In the statistical comparison columns the naming contains 22°C. e.g. p-value(-log10)_LFQ_tig2_22°C_vs_LFQ_Col-0_22°C_S_01_FDR_0.05. Changed to 21°C.

REVIEWERS' COMMENTS

Reviewer #1 (Remarks to the Author):

The authors have done an impressive job of addressing my comments related to the structural aspect of the work thoroughly. The revisions enhance the clarity and depth of the study. Congratulations to the Willmund lab and co-authors on the excellent research and well-presented study!

Reviewer #2 (Remarks to the Author):

I am very pleased with the revision of the manuscript NCOMMS-24-32270A. The authors have performed the requested experiments and addressed all concerns to my full satisfaction. I now support publication in Nature Communications.

Reviewer #3 (Remarks to the Author):

Reviewer #4 (Remarks to the Author):

All points have been addressed to my satisfaction and I recommend publication of the revised article.